# Ube3a unsilencer for the potential treatment of Angelman syndrome

Hanna Vihma [1], Kelin Li[2,8], Anna Welton-Arndt [3,8], Audrey L. Smith [1], Kiran R. Bettadapur [1], Rachel B. Gilmore [4], Eric Gao[1], Justin L. Cotney [4], Hsueh-Cheng Huang[5], Jon L. Collins [6], Stormy J. Chamberlain[4], Hyeong-Min Lee [1,7] ✉, Jeffrey Aubé[2,3] ✉ & Benjamin D. Philpot [1] ✉

Deletion of the maternal *UBE3A* allele causes Angelman syndrome (AS); because paternal *UBE3A* is epigenetically silenced by a long non-coding anti-sense (*UBE3A-ATS*) in neurons, this nearly eliminates UBE3A protein in the brain. Reactivating paternal *UBE3A* holds promise for treating AS. We previously showed topoisomerase inhibitors can reactivate paternal *UBE3A*, but their therapeutic challenges prompted our search for small molecule unsilencers with a different mechanism of action. Here, we found that (*S*)-PHA533533 acts through a novel mechanism to significantly increase paternal *Ube3a* mRNA and UBE3A protein levels while downregulating *Ube3a-ATS* in primary neurons derived from AS model mice. Furthermore, peripheral delivery of (*S*)-PHA533533 in AS model mice induces widespread neuronal UBE3A expression. Finally, we show that (*S*)-PHA533533 unsilences paternal *UBE3A* in AS patient-derived neurons, highlighting its translational potential. Our findings provide a lead for developing a small molecule treatment for AS that could be safe, non-invasively delivered, and capable of brain-wide unsilencing of paternal *UBE3A*.

Angelman syndrome (AS) is a severe neurodevelopmental disorder affecting between 1:10,000 to 1:20,000 births worldwide[1,2]. AS is characterized by ataxia, absence of speech, intellectual disability, and epilepsy[3–5]. Despite these challenges, individuals with AS can live a full lifespan. AS is caused by maternal allele mutations or deletions in the ubiquitin protein ligase E3A (*UBE3A*) gene, which exhibits parent-of-origin genomic imprinting[6–8]. *UBE3A* is known to be biallelically expressed in most cell types except mature neurons, in which the paternally inherited copy of *UBE3A* is intact but epigenetically silenced by a long non-coding antisense RNA (*UBE3A-ATS*)[9]. Since only the maternal copy of *UBE3A* is expressed in mature neurons, maternally

inherited point mutations that impair the E3 ubiquitin ligase function of UBE3A protein are sufficient to produce the full spectrum of AS[8]. Although the genetic cause of AS is well known, there are currently no approved therapies to treat the core symptoms of AS.

The unique epigenetics of *UBE3A* offer a potential therapeutic avenue. While the paternal allele of *UBE3A* is silenced in neurons, it has the capacity to be functionally expressed. Huang et al. first demonstrated this possibility by showing that topoisomerase inhibitors could activate the dormant paternal *Ube3a* allele in mouse neurons[10]. Unfortunately, due to inherent toxicities and poor central nervous system (CNS) penetration of topoisomerase inhibitors, these

[1]Department of Cell Biology and Physiology, Neuroscience Center, and Carolina Institute for Developmental Disabilities, University of North Carolina at Chapel Hill, Chapel Hill, NC, USA. [2]Division of Chemical Biology and Medicinal Chemistry, UNC Eshelman School of Pharmacy, University of North Carolina at Chapel Hill, Chapel Hill, NC, USA. [3]Department of Chemistry, University of North Carolina at Chapel Hill, Chapel Hill, NC, USA. [4]Department of Genetics and Genome Sciences, University of Connecticut School of Medicine, Farmington, CT, USA. [5]Deerfield Discovery and Development, Deerfield Management, New York, NY, USA. [6]Office of the Vice Chancellor for Research, University of North Carolina at Chapel Hill, Chapel Hill, NC, USA. [7]Present address: Department of Biochemistry and Molecular Biology, Hollings Cancer Center, Medical University of South Carolina, Charleston, SC, USA. [8]These authors contributed equally: Kelin Li, Anna Welton-Arndt. ✉e-mail: dr.hmlee@gmail.com; jaube@email.unc.edu; bphilpot@med.unc.edu

compounds could not be advanced as therapeutics for AS[10,11]. However, this discovery inspired further interest in exploring the unsilencing of paternal *Ube3a* as a treatment for AS. Towards this goal, Meng et al. demonstrated that unsilencing of paternal *Ube3a* by inserting a premature termination cassette into *Ube3a-ATS* improves phenotypic defects in AS model mice[12]. Subsequently, paternal unsilencing of *Ube3a* has been achieved by disrupting *Ube3a-ATS* using antisense oligonucleotides (ASOs)[13–17], CRISPR-Cas9 gene therapy[18,19], artificial transcription factors[20], or RNA targeting CRISPR-Cas13[21]. Of these approaches, ASOs are currently in clinical trials for the treatment of AS (clinical trial identifiers: NCT04259281, NCT04428281, NCT05127226). Although ASO treatment strategies for CNS disorders are promising, they also have some serious drawbacks. Due to the short half-life of ASOs and their inability to cross the blood-brain barrier, ASO delivery requires repeated, highly invasive routes of delivery, such as intrathecal (i.t.) or intracerebroventricular (i.c.v.) administration[22]. Mild to severe adverse events of unknown causes can occur in response to ASO administration[23], creating a significant risk associated with applying these otherwise promising therapies for AS. Thus, there remains a critical unmet need to identify safer therapeutics that can be non-invasively delivered to produce brain-wide *UBE3A* unsilencing.

In our efforts to discover and develop new potential AS therapeutics, we performed a high-content screen of a Pfizer chemogenetic library, which led to the discovery of (*S*)-PHA533533, a compound previously developed as a cyclin-dependent kinase 2 (CDK2) inhibitor[24], as a novel small molecule unsilencer of paternal *Ube3a* in mouse neurons. We tested several known analogs of (*S*)-PHA533533 and identified four additional analogs capable of unsilencing paternal *Ube3a*, gaining preliminary insight into the structure-activity relationship (SAR). Notably, unsilencing activity is dependent upon the absolute configuration of (*S*)-PHA533533, with the (*R*)-enantiomer demonstrating a significant loss in activity. We show that (*S*)-PHA533533 downregulates *Ube3a-ATS* and thereby produces *Ube3a* mRNA and UBE3A protein from the paternal allele in maternally *Ube3a*-deficient neurons. Although (*S*)-PHA533533 is known to potently inhibit CDK2 and CDK5[24], we show that the unsilencing of paternal *Ube3a* by (*S*)-PHA533533 is independent of these known targets. Furthermore, we demonstrate that (*S*)-PHA533533 also does not share an unsilencing mechanism with topotecan, as (*S*)-PHA533533 does not broadly inhibit transcriptional elongation. Importantly, we show that (*S*)-PHA533533 can produce widespread unsilencing of *Ube3a* when delivered with a single intraperitoneal (i.p.) injection in AS model mice. Finally, to demonstrate the clinical relevance, we show that (*S*)-PHA533533 effectively increases paternal *UBE3A* mRNA in induced pluripotent stem cell (iPSC)-derived neurons from individuals with AS. These results thus identify a tool compound that can be non-invasively delivered to produce brain-wide paternal *Ube3a* unsilencing and, hence, provide a pathway towards developing a transformative treatment for AS.

## Results

### Identifying (*S*)-PHA533533 as a novel paternal *Ube3a* unsilencer

We aimed to identify novel unsilencers of paternal *Ube3a* by employing a previously developed high-content small molecule screen[10]. This screen was performed in primary neurons derived from knock-in mice harboring a yellow fluorescent protein (YFP) fused to the paternal allele of *Ube3a*[25], allowing for fluorescence-based identification of the paternal allele-specific expression of UBE3A-YFP (detected by a GFP antibody enhancement) (Fig. 1a and Supplementary Table 1). We screened >2800 compounds from a Pfizer chemogenetic library, which contains small molecules selected to cover a broad spectrum of pharmacological space[26]. We screened small molecules at 1 μM concentration and used 0.3 μM topotecan, a topoisomerase inhibitor that can downregulate the *Ube3a-ATS* and unsilence the paternal *Ube3a*, as a positive control (Fig. 1b)[10]. Small molecules that increase the UBE3A-YFP signal represent putative paternal *Ube3a* unsilencers. Because our

neuronal cultures contain a small population of glial cells and immature neurons expressing paternal *Ube3a-YFP*, we also observed small molecules that could decrease the paternal UBE3A-YFP signal—such compounds could be acting by decreasing paternal UBE3A levels in glial cells, decreasing paternal UBE3A levels in immature neurons, and/or inducing cytotoxicity.

Our screen identified (*S*)-PHA533533 ((*S*)-**1** in Supplementary Fig. 1), a compound previously developed as a cyclin-dependent kinase 2 (CDK2) inhibitor[24], as a small molecule that can potently unsilence the paternal *Ube3a-YFP* allele (Fig. 1b). While there was a minimal UBE3A-YFP immunofluorescent signal in our neuronal cultures treated with 0.1% DMSO (vehicle control) or 1 μM (*R*)-PHA533533 (compound (*R*)-**1** in Supplementary Fig. 1; also previously reported[24]) due to the presence of a small population of glial cells and immature neurons that express *Ube3a* biallelically, both 0.3 μM topotecan and 1 μM (*S*)-PHA533533 produced high levels of paternal UBE3A-YFP (Fig. 1c), suggesting that the (*S*)-configuration of **1** is essential for its unsilencing activity. Importantly, (*S*)-PHA533533 is not a structural analog of the previously identified paternal *Ube3a* unsilencer topotecan or other topoisomerase inhibitors (Fig. 1d)[10], suggesting that (*S*)-PHA533533 belongs to a novel class of paternal *Ube3a* unsilencers.

We sought to identify additional active analogs of (*S*)-PHA533533 with the aim of recognizing essential SAR. Accordingly, we tested a selection of seventeen analogs of (*S*)-PHA533533 (Supplementary Fig. 1) for their capability to unsilence paternal *Ube3a* in mouse primary neurons. Of these, we identified four analogs, compounds **2**, **3**, (±)-**4**, and (*S,R*)-**5**, bearing replacement or substitution of the γ-lactam that could unsilence paternal *Ube3a* (Fig. 2), while no activity up to 5 μM concentration was observed in thirteen other tested analogs (compounds **6**–**18** in Supplementary Fig. 1). We established the pharmacological profiles of these unsilencers by determining their cytotoxic concentration ($CC_{50}$, concentration required for the reduction of cell population by 50%), effective concentration ($EC_{50}$, half maximal effective concentration) and maximum effect ($E_{MAX}$, the percent neurons expressing paternal UBE3A-YFP signal at the highest concentration where at least 90% of neurons were alive compared to the DMSO control), and compared these values to topotecan (Fig. 2; see "Methods" for details). Our dose-response analysis indicates that although the set of analogs tested contained molecules with varying potency as CDK2 inhibitors, there was no clear relationship between CDK2 inhibition potency and unsilencing behavior[24,27,28]. Notably, all truncated compounds lacking the α-position methylene linker were inactive as unsilencers even though many of these compounds inhibit CDK2 with an $IC_{50}$ below 1 μM[24,27,28]. In fact, the only changes tolerated to the scaffold of (*S*)-PHA533533 without a substantial loss in unsilencing activity were the replacement of the γ-lactam with select aromatic or heterocyclic moieties, like a naphthyl group (compound **2**), phenyl group (compound **3**), or pyrrolidine group (compound (±)-**4**) (Fig. 2 and Supplementary Fig. 1). Removal of the methyl group appeared to be tolerated as well. Finally, oxidation at the 3-position of the γ-lactam ((*S,R*)-**5**) was tolerated albeit with a marked loss in potency, suggesting that oxidative metabolites of (*S*)-PHA533533 may promote unsilencing. These data suggest that these unsilencers are drug-like small molecules, amenable to further SAR studies.

### (*S*)-PHA533533 and its analogs downregulate *Ube3a-ATS* transcript and unsilence paternal *Ube3a* in primary neurons derived from AS model mice (*Ube3a*^*m−/p+*)

In mouse neurons, the paternal allele of *Ube3a* is epigenetically silenced by a long non-coding antisense transcript (*Ube3a-ATS*)[29–31]. *Ube3a-ATS* is the most distal part of the small nucleolar RNA host gene 14 (*Snhg14*) that is expressed in the opposite orientation of *Ube3a* (Fig. 3a). The proximal part of *Snhg14*, which encodes protein-coding *Snurf/Snrpn*, is transcribed in all tissues, whereas the distal part of *Snhg14*, which among other small

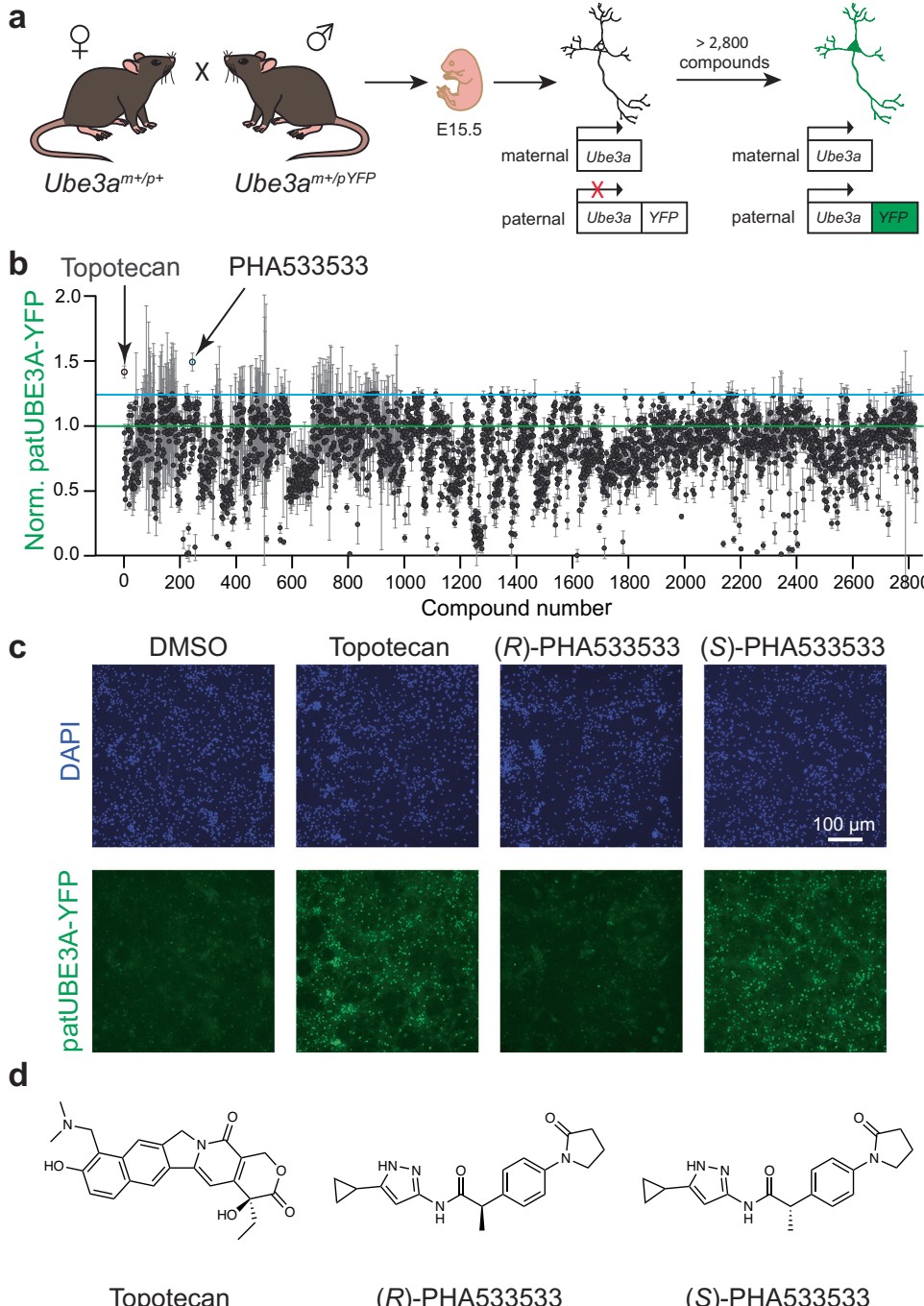

**Fig. 1 | A small molecule screen in primary neurons reveals (*S*)-PHA533533 as a novel paternal *Ube3a* unsilencer. a** Overview of the screening pipeline. Knock-in mice harboring *Ube3a* fused to a yellow fluorescent protein (YFP) allowed visualization of paternal allele-specific expression of UBE3A-YFP (detected by GFP antibody enhancement) in cultured mouse primary neurons. **b** Normalized mean (±SEM) fluorescence intensity of paternal UBE3A-YFP (patUBE3A-YFP) in fixed immunolabeled cells after treating cultured neurons with compounds (@1 µM) from a Pfizer chemogenetic library (>2800 compounds) for 72 h, using 0.3 µM topotecan as a positive control for paternal *Ube3a* unsilencing (single screen, run in quadruplicates; mean ± SEM). The screen revealed (*S*)-PHA533533 as a novel active compound surpassing a 1.25-fold set threshold (cyan line) for potential *Ube3a* unsilencers over DMSO control (green line). **c** Immunofluorescent images of fixed cells demonstrating enantiomer-specific unsilencing of paternal *Ube3a* by PHA533533. Shown are DAPI (nuclear stain) and paternal UBE3A-YFP immunofluorescence in cultured neurons treated with 0.1% DMSO (vehicle), 0.3 µM topotecan (positive control), 1 µM (*R*)-PHA533533 (inactive enantiomer), or (*S*)-PHA533533 (active enantiomer). These experiments have been conducted repeatedly and have consistently yielded similar results. (Scale bar = 100 micrometers). **d** Chemical structures of topotecan, (*R*)- and (*S*)-PHA533533. µM micromolar, DAPI 4′,6-diamidino-2-phenylindole, DMSO dimethyl sulfoxide, E15.5 embryonic day 15.5, GFP green fluorescent protein, Norm. patUBE3A-YFP normalized paternal UBE3A fused to yellow fluorescent protein, SEM standard error of the mean, YFP yellow fluorescent protein. Source data are provided as a Source Data file.

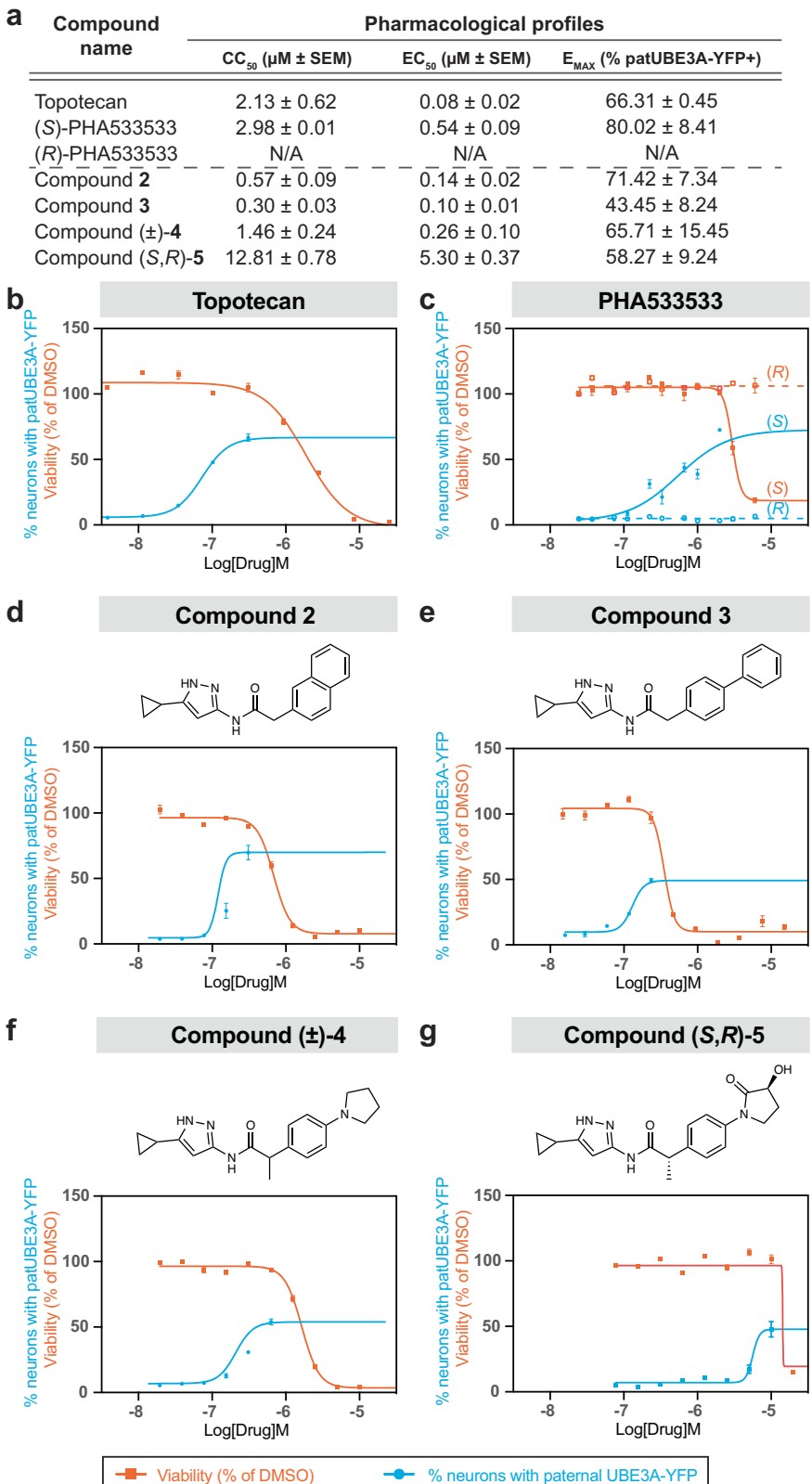

**a**

| Compound name | Pharmacological profiles | | |
|---|---|---|---|
| | CC$_{50}$ (μM ± SEM) | EC$_{50}$ (μM ± SEM) | E$_{MAX}$ (% patUBE3A-YFP+) |
| Topotecan | 2.13 ± 0.62 | 0.08 ± 0.02 | 66.31 ± 0.45 |
| (S)-PHA533533 | 2.98 ± 0.01 | 0.54 ± 0.09 | 80.02 ± 8.41 |
| (R)-PHA533533 | N/A | N/A | N/A |
| Compound **2** | 0.57 ± 0.09 | 0.14 ± 0.02 | 71.42 ± 7.34 |
| Compound **3** | 0.30 ± 0.03 | 0.10 ± 0.01 | 43.45 ± 8.24 |
| Compound (±)-**4** | 1.46 ± 0.24 | 0.26 ± 0.10 | 65.71 ± 15.45 |
| Compound (S,R)-**5** | 12.81 ± 0.78 | 5.30 ± 0.37 | 58.27 ± 9.24 |

nucleolar RNA (snoRNA) genes encodes an array of intron-embedded C/D box snoRNA genes (SNORDs) *Snord116* and *Snord115*, is only transcribed in neurons (Fig. 3a)[32,33].

We hypothesized that (S)-PHA533533 and analogs downregulate *Ube3a-ATS* in neurons, as we have previously demonstrated for topotecan[10], and thereby enable the transcription of paternal *Ube3a*. To test this hypothesis and to determine whether other genes within the *Snhg14* host gene are also affected, we examined the transcriptional effects of (S)-PHA533533 and analogs in mouse primary neurons derived from AS model mice that do not express functional UBE3A from the maternal allele (*Ube3a^{m-/p+}* mice)[34], but express genes within the *Snhg14* host gene at similar levels to wild-type mice as shown comparatively between 0.1% DMSO-treated wild-type (WT) and *Ube3a^{m-/p+}* neurons (Fig. 3b–g). Accordingly, we treated *Ube3a^{m-/p+}*

**Fig. 2 | In vitro pharmacological profile of (S)-PHA533533 and its analogs for unsilencing paternal *Ube3a-YFP*. a** Summary of in vitro cytotoxicity ($CC_{50}$), potency ($EC_{50}$), and efficacy ($E_{MAX}$) for unsilencing paternal *Ube3a-YFP* in mouse primary neurons treated with topotecan, (S)- and (R)-PHA533533, and active PHA533533 analogs **2**, **3**, (±)-**4**, (S,R)-**5** ($n = 3$, run in quadruplicates; mean ± SEM; N/A not applicable). Representative dose-response curves (mean ± SEM) showing cytotoxicity expressed as % viability compared to DMSO control (orange line) and effectiveness expressed as % neurons with paternal UBE3A-YFP (cyan line) for (**b**) topotecan, (**c**) (S)- and (R)-PHA533533, and representative dose-response curves together with the chemical structure for the PHA533533 analogs (**d**) compound **2**, (**e**) compound **3**, (**f**) compound (±)-**4**, and (**g**) compound (S,R)-**5** (mean ± SEM). All drug treatments were administered for 72 h. Only data points where at least 90% of neurons remained viable compared to the DMSO control wells were included. The dose-response assay was repeated three times for each compound ($n = 3$), except for (R)-PHA533533 ($n = 2$). The summary of the pharmacological parameters (mean ± SEM) based on these experiments is depicted in (**a**). µM micromolar, DMSO dimethyl sulfoxide, Log[Drug]M logarithm (base 10) of a molar concentration of a drug, patUBE3A-YFP paternal UBE3A fused to yellow fluorescent protein, SEM standard error of the mean. Source data are provided as a Source Data file.

neurons with 1 µM (S)-PHA533533, 1 µM (R)-PHA533533, 0.1 µM compound **2**, 0.1 µM compound **3**, 0.3 µM compound (±)-**4**, or 5 µM compound (S,R)-**5** and compared the relative quantities of *Ube3a-ATS*, *Snord115*, *Snord116*, *Snrpn*, and *Ube3a* (mRNA) and UBE3A (protein) to those measured from *Ube3a^{m-/p+}* neurons treated with 0.1% DMSO (Fig. 3b–f). The concentrations for the (S)-PHA533533 and (R)-PHA533533 were selected based on the $EC_{100}$ value of the (S)-PHA533533 to detect the maximum effect of (S)-PHA533533, while the concentrations for the other active analogs were selected based on their $EC_{50}$ values to minimize potentially confounding cytotoxic effects. Thus, the effect sizes between treatments with (S)-PHA533533 and its analogs cannot be quantitatively compared by this assay, and all comparisons were made to DMSO vehicle controls.

Our results showed that treatment with (S)-PHA533533 and active analogs **2, 3**, (±)-**4**, and (S,R)-**5** downregulated the *Ube3a-ATS* similarly to topotecan, albeit the reduction for compound **2** did not reach statistical significance, likely due to an apparent outlier experiment (Fig. 3b). The levels of *Ube3a-ATS* from (R)-PHA533533-treated neurons did not differ from those measured from DMSO-treated neurons (Fig. 3b). The observed downregulation of *Ube3a-ATS* led us to test whether these unsilencers affected the expression of other genes upstream of the *Ube3a-ATS* within the *Snhg14* host gene. *Snord115* levels were also significantly reduced by (S)-PHA533533 and topotecan, but its levels were not significantly affected by the treatments with (R)-PHA533533 or the four active analogs (Fig. 3c). The levels of the more upstream genes within the *Snhg14* host gene, such as *Snord116* (Fig. 3d) and *Snrpn* (Fig. 3e), were not affected by treatments with (S)-PHA533533 or any of its active analogs compared to DMSO-treated cells. Only topotecan treatment significantly reduced the levels of *Snrpn*, without affecting *Snord116* levels, which could be potentially explained by a faster processing rate of *Snord116* (Fig. 3e)[16].

Finally, we quantified the extent to which (S)-PHA533533 and its active analogs can produce *Ube3a* mRNA and UBE3A protein levels in *Ube3a^{m-/p+}* neurons that lack maternal *Ube3a* expression. Our results showed that, like topotecan, (S)-PHA533533 and its four active analogs significantly increased *Ube3a* mRNA levels, approaching WT levels in (S)-PHA533533-treated neurons (Fig. 3f). Note that the low but detectable *Ube3a* levels in DMSO and (R)-PHA533533-treated *Ube3a^{m-/p+}* cells likely arise from the small population of glial cells and immature neurons, which biallelically express *Ube3a*, present in our neuronal cultures[6,12,35]. Importantly, UBE3A protein levels in *Ube3a^{m-/p+}* neurons significantly increased compared to background signal from DMSO-treated cells after treatment with topotecan, (S)-PHA533533, compound **3**, or compound (±)-**4** (Fig. 3g). To demonstrate that the unsilencing of paternal *Ube3a* in neurons is independent of the presence of YFP reporter (*Ube3a^{m+/pYFP}*) or the mutation in the maternal *Ube3a* allele (*Ube3a^{m-/p+}*), we treated WT neurons with (S)-PHA533533 and topotecan. As expected, both (S)-PHA533533 and topotecan also significantly increased *Ube3a* mRNA and protein levels in WT neurons (Supplementary Fig. 2). Taken together, these data indicate that the downregulation of *Ube3a-ATS* in mouse primary neurons by (S)-PHA533533 and its active analogs likely mediates the unsilencing of paternal *Ube3a*, whereas the inability of (R)-

PHA533533 to do so demonstrates substantially different biological activity in the (S)-PHA533533 and (R)-PHA533533 enantiomers in unsilencing paternal *Ube3a*.

## The unsilencing of paternal *Ube3a* by (S)-PHA533533 occurs independently from its known CDK2/CKD5 targets and topoisomerase I inhibition

We sought to investigate potential molecular mechanisms by which (S)-PHA533533 unsilences paternal *Ube3a*. (S)-PHA533533 is known to be a potent inhibitor of cyclin-dependent kinases 2 and 5 (CDK2 and CDK5, respectively)[24], therefore we wanted to know whether knocking down these kinases individually using gene-specific ASOs in mouse neurons would produce unsilencing of paternal *Ube3a* or occlude the unsilencing by (S)-PHA533533 treatment (Fig. 4a–d). To test this idea, we first added a gene-specific ASO to our neuronal cultures for two days at 5 days in vitro (DIV5) to achieve a maximum knockdown of our target before treating the neurons with topotecan, (S)-PHA533533, or DMSO as a vehicle for 72 h. Scrambled ASO together with DMSO treatment were used as controls for any non-specific effects. Western blot analysis showed that, although we were able to significantly reduce the levels of CDK2 (-70%, Fig. 4b) and CDK5 (-90%, Fig. 4d) in cultured neurons, this knockdown failed to increase paternal UBE3A-YFP levels compared to DMSO-treated neurons (Fig. 4a, c, respectively). Notably, *Cdk2* levels are very low in the mouse brain compared to *Cdk5*, which made it challenging to detect measurable CDK2 levels in our neuronal cultures and could explain the more modest knockdown for CDK2 than for CDK5. We found that ASO-mediated knockdown of CDK2 or CDK5 failed to unsilence paternal UBE3A-YFP, nor did it occlude the unsilencing by (S)-PHA533533 (Fig. 4a–d).

Next, we examined whether (S)-PHA533533 shares any mechanistic similarity with topotecan in unsilencing paternal *Ube3a*. It has been previously shown that topotecan forms a complex with topoisomerase I (TOP1) and DNA, known as TOP1 cleavage complex (TOP1cc), that is required for the unsilencing of paternal *Ube3a* by topotecan in neurons[36]. Therefore, we tested whether unsilencing by (S)-PHA533533 might occur via an off-target effect on TOP1. For that, we first knocked down TOP1 expression up to 90% using a *Top1*-specific ASO (Fig. 4e, f), which by itself did not result in unsilencing of paternal *Ube3a*, as has been demonstrated previously (Fig. 4e)[36]. TOP1 depletion almost completely occluded the ability of topotecan to unsilence paternal *Ube3a* (Fig. 4e), as expected[36], but did not affect paternal *Ube3a* unsilencing by (S)-PHA533533 treatment (Fig. 4e), suggesting that TOP1 is not the target for (S)-PHA533533. We further confirmed that (S)-PHA533533 is not a TOP1 target by performing a TOP1-mediated DNA relaxation assay. Since relaxed DNA migrates with slower mobility through an agarose gel than supercoiled DNA, we evaluated whether increasing concentrations of (S)-PHA533533 and (R)-PHA533533, or topotecan as a positive control, can inhibit the catalytic activity of TOP1 to relax supercoiled DNA compared to DMSO control. Our results showed that the addition of 3 µM topotecan clearly inhibited TOP1 catalytic activity, as expected, whereas (S)-PHA533533 or (R)-PHA53353 had no effect on the TOP1 activity, demonstrating that (S)-PHA533533 is not targeting TOP1 (Fig. 4g). Lastly, we tested whether, like topotecan, (S)-PHA533533 treatment might reduce the

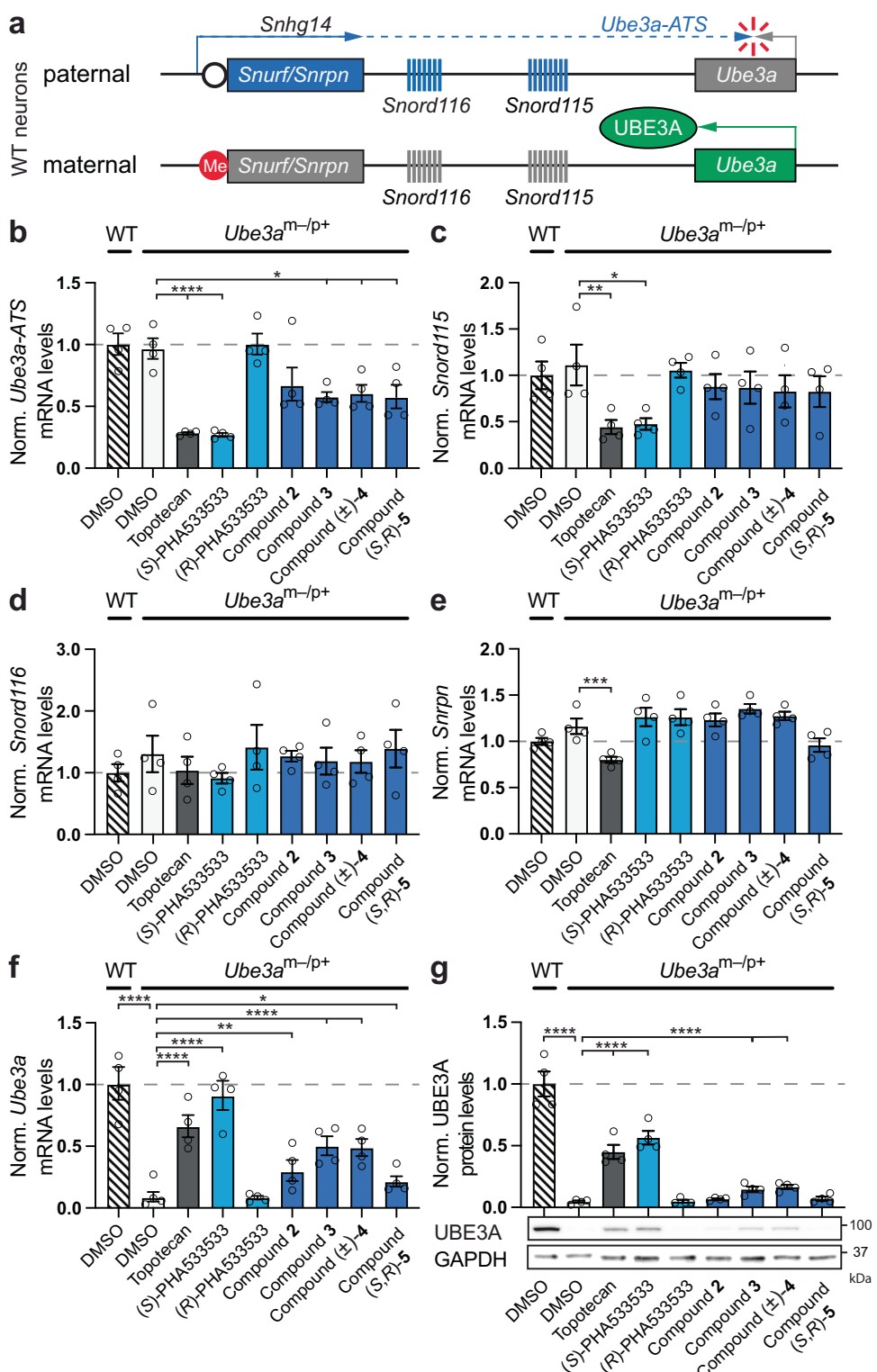

expression of other long genes in addition to *Ube3a-ATS*, since it has been shown that topotecan treatment unsilences paternal *Ube3a* by impairing the transcriptional elongation of nearly all genes longer than 200 kilobases[37]. We treated cultured mouse neurons with either topotecan, (*S*)-PHA533533, or DMSO (vehicle) and compared the relative mRNA levels of *Snord115* (a proxy for *Ube3a-ATS*, Fig. 4h), *Dlg2* (Fig. 4i), and *Nrxn3* (Fig. 4j), as these genes are ≥1 megabase, as well as *Malat1*, one of the longest lncRNAs (Fig. 4k). As expected, topotecan significantly impaired the expression of all the measured long genes

(Fig. 4h–k), whereas (*S*)-PHA533533-treatment significantly reduced the expression of only *Snord115* (Fig. 4h). To understand whether (*S*)-PHA533533 affects the expression of other known antisense lncRNA transcripts, we also measured the expression of *Pantr1* (Fig. 4l), an intergenic antisense lncRNA that shares a bidirectional promoter with *Pou3f3* gene, and *Airn* (Fig. 4m), a paternally expressed antisense of *Igf2r* gene[38]. Our results showed that neither topotecan nor (*S*)-PHA533533 treatments changed the expression levels on *Pantr1* or *Airn*. Taken together, our data indicate that (*S*)-PHA533533 is unlikely

**Fig. 3 | (S)-PHA533533 and its analogs downregulate the *Ube3a* antisense (*Ube3a-ATS*) transcript and produce UBE3A protein from the paternal allele in mouse primary neurons derived from Angelman syndrome model mice.**
**a** Simplified schematics of *Ube3a* and *Ube3a-ATS* in mature mouse neurons. *Ube3a-ATS* transcript, which together with *Snurf/Snrpn*, *Snord107* (not depicted), *Snord64* (not depicted), *Snord116*, and *Snord115* is part of the *Snhg14* host gene, is expressed only from the paternal allele (blue dashed line), where it represses the expression of *Ube3a*. Protein coding *Snurf*/Snrpn is expressed in all tissues (solid blue line), whereas other paternally expressed genes are exclusively expressed in neurons (blue dashed line). Maternal *Ube3a-ATS* expression is epigenetically silenced due to the methylation stretch of CpG islands at the *Snrpn* promoter region (red circle) that is unmethylated on the paternal allele (white circle), resulting in monoallelic expression of *Ube3a*/UBE3A in neurons (green arrow/ellipse). **b–g** Primary cortical neurons derived from wild-type (WT) or Angelman syndrome model (*Ube3a^{m−/p+}*)

mice were treated either with 0.1% DMSO as a vehicle, 0.3 μM topotecan, 1 μM (*S*)- or (*R*)-PHA533533, 0.1 μM compound **2**, 0.1 μM compound **3**, 0.3 μM compound (±)-**4**, or 5 μM (*S,R*)-**5** as indicated for 72 h at DIV7. Following the treatment, the relative quantities of (**b**) *Ube3a-ATS*, (**c**) *Snord115*, (**d**) *Snord116*, (**e**) *Snrpn*, and (**f**) *Ube3a* transcripts were determined by quantitative RT-PCR and the expression of (**g**) UBE3A protein was determined by western blotting. All data were normalized to GAPDH/*Gapdh* or β-ACTIN expression, log-transformed, mean-centered, and autoscaled for statistical analysis (*n* = 4 experiments; mean ± SEM, one-way ANOVA followed by Dunnett's post hoc test, **p* < 0.05, ***p* < 0.01, ****p* < 0.001, *****p* < 0.0001, NS non-significant). DMSO dimethyl sulfoxide, kDa kilodalton, Norm. normalized, me methylation. Source data and comprehensive statistics, including *F*-values, degrees of freedom, confidence intervals, and exact *p* values, are provided as a Source Data file.

to unsilence *Ube3a* via its known mechanism of CDK2/5 inhibition, through topoisomerase inhibition, or through a general effect on the expression of antisense transcripts, suggesting that (*S*)-PHA533533 instead employs a novel mechanism of paternal *Ube3a* unsilencing.

## Peripheral delivery of (*S*)-PHA533533 can produce brain-wide unsilencing of paternal *Ube3a* in AS model mice

We next examined whether (*S*)-PHA533533 can unsilence paternal *Ube3a* in the mouse brain in vivo and therefore rescue UBE3A expression in AS model mice that otherwise lack neuronal UBE3A[34]. Being a potent inhibitor of CDK2/CDK5, (*S*)-PHA533533 was developed as an anti-tumor drug and has been previously preclinically tested by administrating 7.5 mg/kg orally into a human ovarian A2780 xenograft mouse model twice a day for 20 days without any signs of toxicity[24]. Although (*S*)-PHA533533 is well tolerated in adult mice when delivered systemically, we aimed to administer (*S*)-PHA533533 into much younger mice at postnatal day 11 (P11). For that, we conducted a preliminary dose-response evaluation to determine the optimal dosage for our model. Through systematic testing of different doses, we found that the 2 mg/kg dosage was tolerated in P11 mice. Accordingly, we injected (i.p.) AS model mice with (*S*)-PHA533533 (2 mg/kg) or saline at postnatal day 11 (P11) and compared the relative *Ube3a* mRNA levels in different areas of the brain to those in saline-injected WT and AS mice 24 h post-injection. Our results showed that a single dose of (*S*)-PHA533533 significantly increases the levels of *Ube3a* in AS mice compared to saline-treated control mice in all examined regions of the brain (Fig. 5A). The levels of unsilenced *Ube3a* were most pronounced in the olfactory bulb, where *Ube3a* mRNA levels in (*S*)-PHA533533-treated AS mice reached those measured from WT saline-treated mice (Fig. 5A).

To better evaluate the biodistribution of reinstated UBE3A protein at a cellular resolution in (*S*)-PHA533533-treated AS mice compared to saline-treated control WT and AS mice, we transcardially perfused the treated animals 48 h post-injection for immunohistochemical analysis (Fig. 5B–M). Consistent with our *Ube3a* mRNA results, we found that (*S*)-PHA533533 produced a widespread reinstatement of UBE3A protein levels throughout the AS mouse brain (Fig. 5D). Although paternal UBE3A levels in (*S*)-PHA533533-treated AS mice did not reach the levels of maternal UBE3A in WT control mice (Fig. 5B–D), increased UBE3A signal was detected in all NeuN positive neurons in different areas of the brain, indicating efficient unsilencing of paternal *Ube3a* in mature neurons (Fig. 5E–M). These results serve as proof-of-concept that systemic delivery of (*S*)-PHA533533 can produce widespread neuronal expression of UBE3A protein throughout the AS model mouse brain.

## (*S*)-PHA533533 downregulates the *UBE3A* antisense (*UBE3A-ATS*) transcript and increases paternal *UBE3A* mRNA in neurons derived from patients with Angelman syndrome

To test whether (*S*)-PHA533533 has translational potential to unsilence *UBE3A* in human neurons, we used cultures composed primarily of

forebrain cortical neurons derived from human iPSCs carrying an ~5.5 Mb deletion of the maternally-inherited allele of chromosome 15q11-q13. This line derived from a patient with AS (del1-0), also contains an additional 24 kb bipartite boundary element region deletion, encompassing IPW and PWAR1 of the paternally inherited allele (ΔI-P), which results in imprinting of *UBE3A* earlier in the differentiation process (Fig. 6a)[31,39]. It is important to note that while the imprinting of *UBE3A* in neurons lacking the boundary element is expedited compared to cell lines containing the boundary element, the level at which *UBE3A* becomes imprinted in culture is also influenced by factors such as sufficient expression of *UBE3A-ATS/SNRPN*, cellular heterogeneity, and culture maturity[31]. Our results showed that (*S*)-PHA533533-treatment significantly increased *UBE3A* levels (Fig. 6b) and decreased *UBE3A-ATS* levels (Fig. 6c) compared to DMSO-treated neurons in human 6.5-week-old ASdel1-0ΔI-P neurons, demonstrating that (*S*)-PHA533533 can unsilence paternal *UBE3A* in AS patient-derived neurons. Moreover, our findings illustrate that treatment with (*S*)-PHA533533 increases UBE3A protein levels in mature iPSC-derived human neurons from an Angelman syndrome patient (ASdel1-0 neurons), as confirmed through western blot and immunocytochemistry assays (Supplementary Fig. 3). While (*S*)-PHA533533 treatment increases *UBE3A* mRNA and UBE3A protein levels in neurons, it does not affect the levels of *UBE3A*/UBE3A in cells that express *UBE3A* biallelically, such as HEK293T cells (Supplementary Fig. 4).

Next, we established the pharmacological profiles for (*S*)-PHA533533 and (*R*)-PHA533533 in human ASdel1-0ΔI-P neurons by determining their cytotoxic concentration (CC$_{50}$) based on a viability assessment using Ct values of a housekeeping gene *GAPDH*, the effective concentration (EC$_{50}$) for unsilencing paternal *UBE3A*, and inhibitory concentration (IC$_{50}$) for reducing the expression on *UBE3A-ATS* (Fig. 6d). The dose-response analysis revealed a mean CC$_{50}$ for (*S*)-PHA533533 in human neurons to be ~3.7 μM (Fig. 6d, e). The mean EC$_{50}$ value was slightly higher than the IC$_{50}$ value for (*S*)-PHA533533, being ~0.59 μM and ~0.42 μM, respectively (Fig. 6d, f). Similar to our results from mouse neurons, (*R*)-PHA533533 was unable to produce unsilencing of paternal *UBE3A* in ASdel1-0ΔI-P neurons (Fig. 6d–f), demonstrating that PHA533533 produces enantiomer-specific unsilencing also in human neurons. Taken together, these results show that (*S*)-PHA533533 is highly effective at unsilencing paternal *UBE3A* in AS patient-derived neurons, demonstrating the clinical relevance of our novel unsilencer.

## Discussion

The dormant paternal *UBE3A* allele presents as a unique therapeutic target for AS. Here we performed a high-content screen to identify novel small molecule unsilencers of paternal *UBE3A*. We discovered (*S*)-PHA533533, which effectively downregulates *Ube3a-ATS/UBE3A-ATS* and thereby epigenetically unsilences paternal *Ube3a/UBE3A* in both mouse and human neurons. We further identified four additional analogs of (*S*)-PHA533533 capable of unsilencing paternal *Ube3a*,

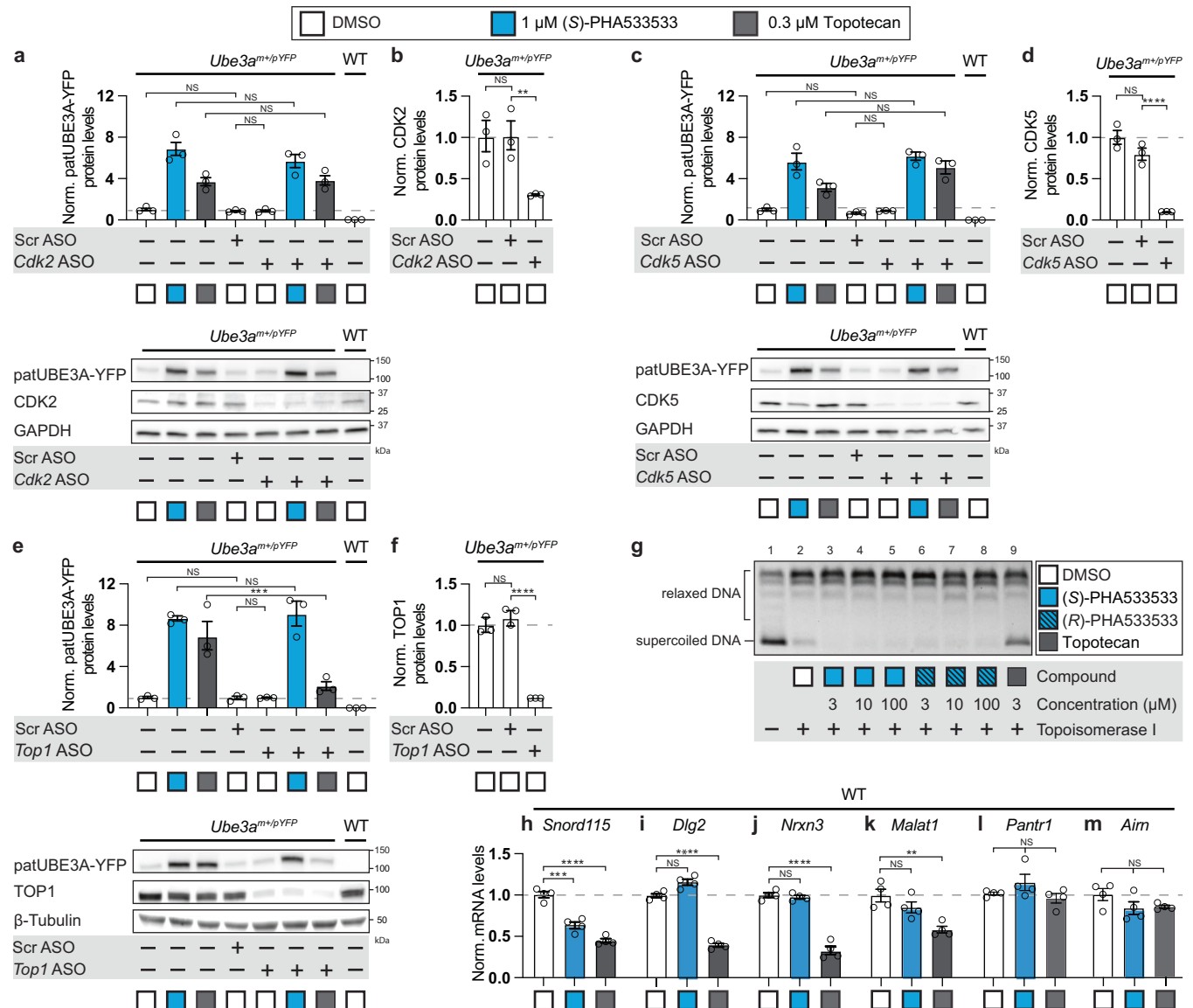

**Fig. 4 | (S)-PHA533533 unsilences paternal *Ube3a* via a mechanism that is independent of CDK2/CDK5 and topoisomerase 1 (TOP1) inhibition.** Quantification and representative western blots showing that ASO-mediated knock-down of (**a**, **b**) CDK2, (**c**, **d**) CDK5, or (**e**, **f**) TOP1 failed to unsilence paternal *Ube3a-YFP* or occlude the unsilencing of paternal *Ube3a-YFP* by (S)-PHA533533 in primary cortical neurons, whereas (**e**, **f**) ASO-mediated knock-down of TOP1 occluded the TOP1 inhibitor topotecan from unsilencing paternal *Ube3a-YFP*. For these experiments, mouse primary neurons derived from wild-type or paternal *Ube3a-YFP* reporter mice (*Ube3a^{m+/pYFP}*) were treated as indicated with 10 μM active or scrambled (Scr) ASOs at DIV5 followed by treatment with 0.1% DMSO (vehicle control), 1 μM (S)-PHA533533, or 0.3 μM topotecan for 72 h starting at DIV7 (*n* = 3 experiments; mean ± SEM). **g** TOP1-based DNA relaxation assay verified that (S)-PHA533533 and (R)-PHA533533 do not target TOP1. The addition of TOP1 catalyzes supercoiled DNA substrate (lane 1) into slower migrating relaxed DNA (lane 2). The presence of different concentrations of (S)-PHA533533 (lanes 3–5) or (R)-PHA533533 (lanes 6–8)

did not result in an increased amount of uncut supercoiled DNA compared to DMSO control (lane 2), whereas the presence of 3 μM topotecan did (lane 9). Data shown are representative of multiple independent trials, all confirming similar outcomes. **h–j** (S)-PHA533533 reduced the expression of *Ube3a-ATS*, but not other long genes or other antisense transcripts. Following the treatment, the relative quantities of (**h**) *Snord115*, (**i**) *Dlg2*, (**j**) *Nrxn3*, (**k**) *Malat1*, (**l**) *Pantr1*, and (**m**) *Airn* transcripts were determined by quantitative RT-PCR (*n* = 4 experiments; mean ± SEM). All data in (**a**–**f**) and (**h**–**m**) were normalized to GAPDH/*Gapdh* or β-TUBULIN expression, log-transformed, mean-centered, and autoscaled for statistical analysis (one-way ANOVA followed by Tukey's post hoc test, \**p* < 0.05, \*\**p* < 0.01, \*\*\**p* < 0.001, \*\*\*\**p* < 0.0001, NS non-significant). μM micromolar, DMSO dimethyl sulfoxide, kDa kilodalton, patUBE3A-YFP paternal UBE3A fused to yellow fluorescent protein, YFP yellow fluorescent protein, WT wild-type. Source data and comprehensive statistics, including *F*-values, degrees of freedom, confidence intervals, and exact *p* values, are provided as a Source Data file.

gaining preliminary insight into the SAR. We demonstrated that paternal *Ube3a/UBE3A* unsilencing by PHA533533 depends on its absolute stereochemistry: only the (S) enantiomer demonstrates notable unsilencing activity. Finally, we established (S)-PHA533533 as the first small molecule capable of brain-wide unsilencing of paternal *Ube3a* in AS model mice upon systemic delivery. Thus, (S)-PHA533533 provides us with a powerful tool to be leveraged toward developing a transformative, noninvasively delivered therapeutic for AS.

AS treatment strategies based on the reinstatement of UBE3A expression typically utilize virus-mediated *UBE3A* gene replacement[40–42], virus-mediated methods of *UBE3A-ATS* disruption[18–21], or ASO-mediated *UBE3A-ATS* knockdown[13–17]. While these approaches can yield significant phenotypic improvements in mouse models, they also carry inherent limitations and risks. Notably, current adeno-associated virus (AAV)- and ASO-based treatment methods require patients to endure general sedation followed by invasive intracranial or intra-cerebrospinal fluid

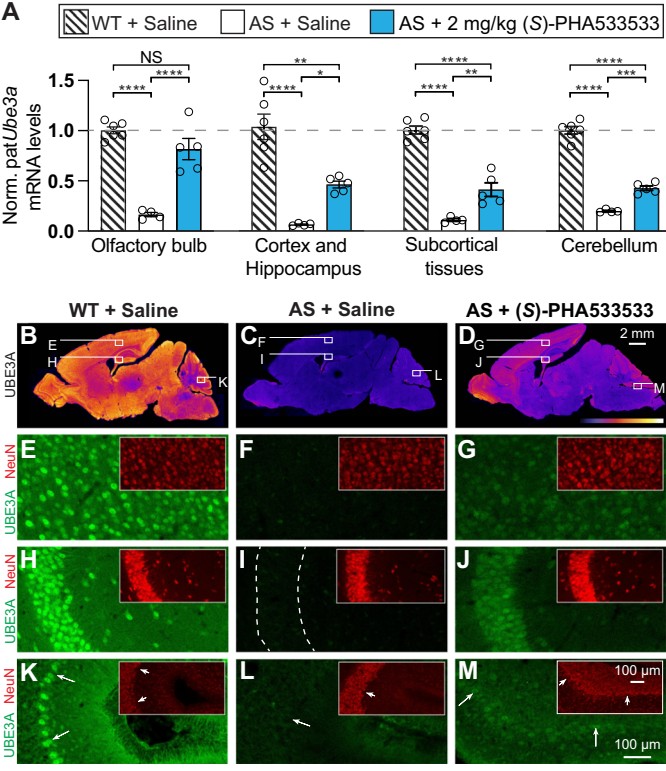

**Fig. 5 | A single peripheral administration of (*S*)-PHA533533 unsilences paternal *Ube3a* across the brain.** Comparison of wild-type (WT) or Angelman syndrome (AS) model mice (*Ube3a^{m−/p+}* mice) given saline vehicle or (*S*)-PHA533533 (2 mg/kg; i.p.) on P11. **A** qPCR measurements of *Ube3a* mRNA from indicated brain regions taken at P12. All data were normalized to *Gapdh* expression. Each data point represents an individual animal (*n* = 6/WT + Saline; *n* = 5/AS + (*S*)-PHA533533; *n* = 4/AS + Saline; mean ± SEM, one-way ANOVA followed by Tukey's post hoc test, **p* < 0.05, ***p* < 0.01, ****p* < 0.001, *****p* < 0.0001, NS non-significant). Low magnification of sagittal brain sections showing immunofluorescence for UBE3A protein colored using a fire gradient in saline-treated (**B**) wild-type mouse, (**C**) AS model mouse, and (**D**) (*S*)-PHA533533-treated AS model mouse at P13 (scale bar = 2 millimeters). Higher magnification view of UBE3A protein in green and NeuN protein in red in (**E**–**G**) cortex, (**H**–**J**) hippocampal CA3 region, and (**K**–**M**) cerebellum (arrows indicate Purkinje cells) as indicated in (**B**–**D**). Experiments in (**B**–**M**) have been conducted repeatedly and have consistently yielded similar results. (Scale bar = 100 micrometers). Norm. normalized. All data and comprehensive statistics, including *F*-values, degrees of freedom, confidence intervals, and exact *p* values, are available in the Source Data file.

injection. Systemic delivery of (*S*)-PHA533533 via i.p. injection, a comparatively less invasive route of administration, produced brain-wide *Ube3a* unsilencing in AS model mice. Future studies of (*S*)-PHA533533 or its active analogs may reveal opportunities for minimally invasive subcutaneous or oral delivery.

A maximally effective AS therapeutic should succeed in reinstating UBE3A across the entire brain, especially during the critical early postnatal period[43], closely mimicking natural UBE3A expression patterns observed in both mouse and non-human primate (NHP) central nervous system (CNS)[42]. Our proof-of-concept in vivo study demonstrates that (*S*)-PHA533533 via i.p. injection unsilences UBE3A expression in neurons throughout the AS model mouse brain (Fig. 5B–M). Importantly, the brain biodistribution of (*S*)-PHA533533-mediated *Ube3a* unsilencing dramatically eclipses that achieved by the only other known class of small molecule *Ube3a* unsilencers—topoisomerase inhibitors—whose bioactivity is proximally limited to CNS sites of delivery[10]. The capacity for AAV-mediated therapies to achieve brain-wide biodistribution is similarly limited, particularly as postnatal development progresses[44]. In contrast, ASO-mediated *UBE3A-ATS*

knockdown leads to widespread paternal *Ube3a/UBE3A* unsilencing in the CNS. Indeed, ASOs administered to NHPs through i.c.v or i.t. routes exhibit widespread brain biodistribution regardless of the age of administration[45], albeit with differences in their activity and duration of action among various cell types and brain regions[46]. Consequently, repeated dosing is necessary to sustain ASO-mediated knockdown of *UBE3A-ATS* and the therapeutic benefits of paternal *UBE3A* unsilencing, resulting in procedural hardship for patients enrolled in ongoing clinical trials (clinical trial identifiers: NCT04259281, NCT04428281, NCT05127226). A peripherally delivered small molecule *UBE3A* unsilencer offers a promising alternative, with benefits including reversibility, titratability, and straightforward and cost-effective manufacturing. Thus, a small molecule treatment for AS might not only offer profound, lifelong benefits as a stand-alone treatment modality for a broad patient demographic, but it could also serve as a complementary treatment alongside other therapeutic strategies in development.

(*S*)-PHA533533 was developed as an optimized drug-like inhibitor of cyclin-dependent kinase 2 (CDK2) to target tumors[24]. It exhibits inhibitory potency against CDK2/cyclin A and CDK2/cyclin E complexes, with $IC_{50}$ values of 37 nM and 55 nM, respectively. Additionally, it shows comparable inhibitory effect against the CDK5/p25 complex with an $IC_{50}$ of 65 nM[24]. Importantly, (*S*)-PHA533533 demonstrated promising pharmacokinetic characteristics both in vitro and in vivo: it has a high Caco-2 cell permeability, exhibited good aqueous solubility (>200 µM), was found to be ~74% plasma protein bound in a high-throughput assay, and is mostly stable when exposed to cytochrome P450 3A4 for 90 minutes (96% remaining)[24]. Following a 10 mg/kg oral dose in CD-1 mice, (*S*)-PHA533533 displayed a clearance rate of 5.8 ml/min/kg and its peak concentration ($C_{max}$) reached 21 µM with an area under the curve (AUC) of 96 µM/h[24]. Given the unsilencing properties of (*S*)-PHA533533 that we identified, coupled with its adaptable chemical structure and established pharmacokinetic properties, we propose that this class of compounds is a prime candidate in our quest for a small molecule unsilencing agent. Surprisingly, (*S*)-PHA533533 does not appear to unsilence paternal *Ube3a* through its known molecular mechanisms, as we were unable to unsilence paternal *Ube3a*, nor occlude the unsilencing ability of (*S*)-PHA533533, by knocking down CDK2 or CDK5 (Fig. 4a–d). In further evidence that (*S*)-PHA533533 does not unsilence paternal *Ube3a* through CDK2/5 inhibition, we did not detect a clear relationship between CDK2 inhibition potency and paternal *Ube3a* unsilencing efficacy for the active analogs. The observation that (*S*)-PHA533533 does not work through its known molecular mechanism led us to hypothesize that it might be working through an off-target effect on topoisomerases, as previous studies have shown that Top1 inhibition can potently unsilence paternal *Ube3a* by impairing the transcriptional elongation of long genes, including *Ube3a-ATS*[10,11,37]. Yet, our results showed no evidence that (*S*)-PHA533533 inhibits either Top1 or the expression of long genes, and it did not reduce the expression of candidate antisense and imprinted lncRNAs besides *Ube3a-ATS* (Fig. 4e–m). Collectively, these data indicate that (*S*)-PHA533533 employs a novel mechanism of action (MOA) distinct from previously identified small molecule *Ube3a* unsilencing agents[10,11]. Importantly, the MOA of (*S*)-PHA533533 appears to be conserved across species, as we detected very similar in vitro $EC_{50}$ values of 0.54 µM and 0.59 µM in mouse and human neurons, respectively. We aim in future work to identify the MOA by which (*S*)-PHA533533 unsilences paternal *UBE3A*. Such an endeavor should be aided by our finding that (*S*)- but not (*R*)-PHA533533 potently unsilences paternal *Ube3a/UBE3A*, as well as by an extensive SAR campaign based on the modular heterocyclic scaffold of the compound that is readily adaptable for exploration in medicinal chemistry. While small molecules can be advanced to the clinic without a known MOA, mechanistic insights would help inform a SAR

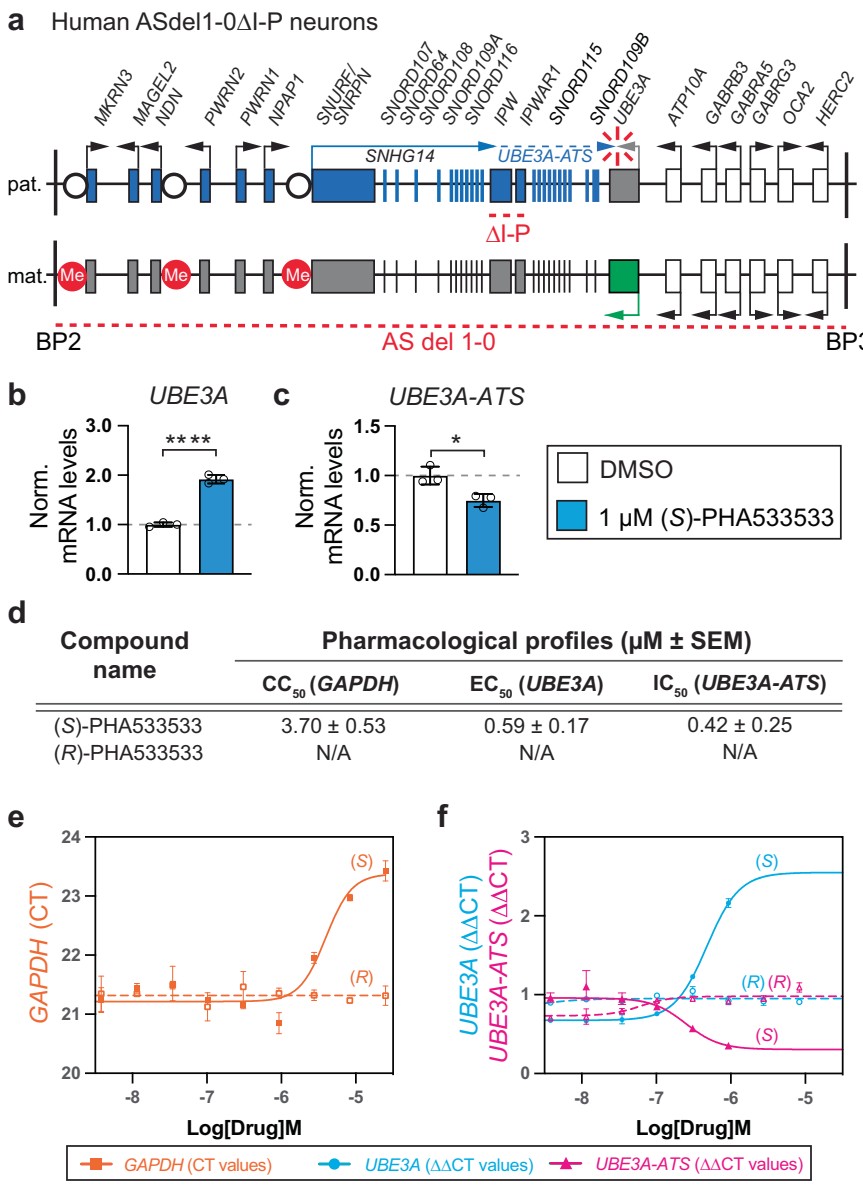

**Fig. 6 | (*S*)-PHA533533 downregulates the *UBE3A* antisense (*UBE3A-ATS*) transcript and increases paternal *UBE3A* mRNA in neurons derived from patients with Angelman syndrome. a** Schematics of human chromosome 15q11–q13 region between breakpoints 2 and 3 (BP2 and BP3) where the deleted regions on the paternal (pat.) and maternal (mat.) alleles in AS patient-derived iPSCs used for neuronal differentiation are denoted by red dotted lines. Blue and green boxes denote genes expressed exclusively from paternal or maternal alleles, respectively. White boxes denote biallelically expressed genes. Differentially unmethylated or methylated regions are depicted with a white circle or red circle, respectively. Arrows indicate the direction of transcription. The proximal part of the *SNGH14* host gene is expressed in most cell types (solid blue line), whereas the distal part of *SNGH14* including *UBE3A-ATS* is exclusively expressed in neurons (dashed blue line). (Modified from Martins-Taylor et al.[50]). Human neurons differentiated from ASdel1-0 ΔI-P iPSCs for 6.5 weeks were treated either with 0.1% DMSO as a vehicle or 1 μM (*S*)-PHA533533 for 72 h. Following the treatment, the relative quantities of (**b**) *UBE3A* and (**c**) *UBE3A-ATS* were determined by quantitative RT-PCR ($n = 3$ experiments, each run in duplicate cultures; mean ± SEM, data were log-transformed and mean-centered for statistical analysis, two-tailed *T*-test, *$p < 0.05$, ****$p < 0.0001$). **d** Summary of in vitro cytotoxicity ($CC_{50}$), based on viability assessment, and potency ($EC_{50}$ for *UBE3A* and $IC_{50}$ for *UBE3A-ATS*) in human ASdel1-0ΔI-P neurons for (*S*)- and (*R*)-PHA533533 ($n = 3$ experiments run in duplicate cultures; mean ± SEM; N/A not applicable. Representative dose-response curves (mean ± SEM from duplicated cultures) for (*S*)-PHA533533 and (*R*)-PHA533533 showing (**e**) cytotoxicity expressed as quantitative RT-PCR cycle threshold (CT) value for *GAPDH* (orange line), and (**f**) effectiveness expressed as ΔΔCT value for *UBE3A* (cyan line) and *UBE3A-ATS* (magenta line). The dose-response assay was repeated three times for each compound ($n = 3$), except for (*R*)-PHA533533 ($n = 1$). The summary of the pharmacological parameters (mean ± SEM) based on these experiments is depicted in (**d**). All data in (**b**–**d**) and (**f**) were normalized to *GAPDH* expression. μM micromolar, DMSO dimethyl sulfoxide, Norm. normalized, SEM standard error of the mean. Source data and comprehensive statistics, including *F*-values, degrees of freedom, confidence intervals, and exact *p* values, are provided as a Source Data file.

campaign intended to improve the efficacy of the compound, as well as predict and limit potential treatment side effects.

While (*S*)-PHA533533 offers an inroad to developing a potentially transformative and non-invasive treatment for AS, and our results here offer an important first step in this process, many unknowns must be first explored preclinically before a clinical trial is warranted. As a first step, we must continue SAR studies of (*S*)-PHA533533 to identify compounds with improved paternal *Ube3a* unsilencing effectiveness in tandem with improving CNS drug-like properties and minimizing off-target effects. Although we already identified four active analogs

and thirteen inactive analogs of (S)-PHA533533, these compounds were originally designed to optimize CDK2/cyclin A inhibition as potential antitumor agents[24], and were not developed with other MOAs in mind. Given that the unsilencing MOA does not involve CDK2 or cyclin inhibition, the previously published SAR is not applicable, prompting a thorough reassessment from the ground up. It is essential to determine how the paternal *Ube3a* unsilencing SAR differs from that already established at CDK2 and if the scaffold can be evolved to improve unsilencing efficacy while mitigating inhibition of the CDK family. Ideally, target identification will aid in this endeavor and help to identify essential ligand-target interactions that promote unsilencing behavior. Secondly, we must establish whether the unsilencing of *Ube3a* by our novel unsilencer occurs at all stages of brain development. For our proof-of-concept in vivo study, we administrated our lead compound (S)-PHA533533 into juvenile mice at P11 which is within the optimal three-week postnatal treatment window for AS mice[43]. However, further research is essential to ascertain whether unsilencing can occur throughout life. Moreover, the duration of the unsilencing effect across developmental stages needs evaluation to establish the optimal dosing regimen. Finally, to advance (S)-PHA533533 or its analogs as a potential therapeutic for AS, a comprehensive safety assessment is imperative. (S)-PHA533533 has been previously evaluated preclinically as an antitumor agent by administrating 7.5 mg/kg orally to an adult mouse model with human ovarian A2780 xenograft twice daily for 20 days without observed toxicity[24], a dosage that exceeds the level found effective for unsilencing paternal *Ube3a* in our preliminary study. However, given the focus on AS therapies for pediatric patients, it is essential to conduct juvenile animal toxicology studies to ascertain the safety of our novel unsilencer during crucial developmental stages[47].

In sum, our discovery of (S)-PHA533533 as an unsilencer of paternal *UBE3A* offers a lead for developing a non-invasive therapy for AS, and we propose that a more extensive drug development campaign around this compound is now fully warranted.

## Methods

### Mice
Knock-in mice expressing yellow fluorescent protein (YFP) fused to the C-terminus of *Ube3a* (Dindot et al.[25]) (RRID:IMSR_JAX:017765) were used for the high-content screen and initial hit validations. Mice with paternal *Ube3a-YFP* inheritance (*Ube3a$^{m+/pYFP}$*) were generated by mating wild-type females to paternal *Ube3a-YFP* (*Ube3a$^{m+/pYFP}$*) males. Angelman syndrome model mice (*Ube3a$^{m-/p+}$*; RRID:IMSR_JAX:016590)[34] were used for additional in vitro and in vivo studies. Maternal *Ube3a*-deficient mice were generated by crossing congenic wild-type males to phenotypically normal paternal *Ube3a*-deficient females (*Ube3a$^{m+/p-}$*)[34]. Embryonic male and female mice were randomly used for cortical neuronal isolation and culture. All mice used in this work were backcrossed to obtain a congenic C57BL/6J genetic background. Primers used for genotyping are listed in Table 1. Mice were housed on a 12:12 light-dark cycle, at a temperature range between 20 and 26 °C, and at optimal humidity to prevent dryness or excessive moisture. Mice had *ad libitum* access to food and water. Cervical dislocation was used to euthanize 3 to 5-month-old timed pregnant female mice for preparing mouse primary neuron cultures. For immunohistochemical assays, mice at P13 were deeply anesthetized with euthasol (100 mg/kg, i.p.) prior to transcardial perfusion. Sex of the mice was not considered as a biological variable. All animal experiments were conducted under IACUC protocols #20-156 and #23-138 approved by the University of North Carolina School of Medicine.

### Chemistry, drug preparation, and topoisomerase I-mediated DNA relaxation assay
The Pfizer chemogenetic library and PHA533533 derivatives were provided by the Compound Transfer Program (CTP) at Pfizer.

### Table 1 | Sequences of primers used for genotyping

| Primer name | Sequence |
| --- | --- |
| Ube3aYFP_Fwd | CACATGAAGCAGCACGACTT |
| Ube3aYFP_Rev | AGTTCACCTTGATGCCGTTC |
| AS_P1 | ACTTCTCAAGGTAAGCTGAGCTTGC |
| AS_P2 | GCTCAAGGTTGTATGCCTTGGTGCT |
| AS_P3 | TGCATCGCATTGTGTGAGTAGGTGTC |

Topotecan hydrochloride salt was purchased from Cayman Chemicals (Item No. 14129). Resynthesis of (S)-PHA533533 and analogs was carried out as previously described[24,27,28]. For in vitro assays, all compounds were pre-dissolved in DMSO, with a final concentration of 0.1% DMSO. For in vivo assays, (S)-PHA533533 was dissolved in 0.9% NaCl. The Human Topoisomerase I Assay Kit was purchased from TopoGEN and performed according to the manufacturer's protocols.

### Mouse primary neuron culture
Cerebral cortices from E15.5 embryos carrying paternal *Ube3a-YFP* (*Ube3a$^{m+/pYFP}$*) or maternal *Ube3a*-deficient mice (*Ube3a$^{m-/p+}$*) were dissected in Leibovitz's L-15 Medium (11415064, ThermoFisher) and rinsed with HBSS (14025076, Thermo Fisher). Cortices were then incubated for 30 min at 37 °C in papain (1 vial diluted with 2.5 ml of HBSS; 88285, ThermoFisher) with DNase I (20 mg/ml; D4513, Sigma) in HBSS. Next, 1 ml of Neurobasal Plus medium (A3582901, Thermo-Fisher) containing 5% FBS (TMS-013-B; MilliporeSigma) was added to the cortical mixture to deactivate the papain and subsequently triturated. The cells were centrifuged for 2 min at $4600 \times g$, washed with HBSS, and resuspended in a complete medium containing Neurobasal Plus medium with 5% fetal bovine serum, GlutaMax (35050-061, Invitrogen), B27 Plus (A3582801, ThermoFisher), and Antibiotic-Antimycotic (15240-062, Invitrogen). Isolated cortical neurons were plated onto poly-D-lysine coated 384-well plates at $2 \times 10^4$ cell/well for screening and dose-response assays, or 6-well plates at $1 \times 10^6$ cell/well for molecular and biochemical assays (western blot, ASO-dependent knock-down, and quantitative RT-PCR). Cultured neurons were maintained by replacing half of the medium containing Neurobasal Plus, GlutaMax, B27 Plus, and 2.46 μg/ml 5-fluoro-2´-deoxyuridine (F0503, Sigma) every 3–4 days.

### Human H9 hESC, ASdel1-0 iPSC and ASdel1-0ΔI-P iPSC cultures and differentiation
The detailed generation, characterization, culture, and differentiation of human ASdel1-0 and ASdel1-0ΔI-P iPSCs lines were previously described in Chamberlain et al.[39] and Hsiao et al.[31], respectively. Briefly, the ASdel1-0ΔI-P cell line is derived from ASdel1-0 iPSCs[39], an AS cell line with a large deletion of maternal chromosome 15q11-q13. They harbor a deletion of a boundary element (IPW/PWAR1) that terminates transcription of *UBE3A-ATS* in non-neurons. The wild-type H9 (WA09) human ES (hESCs) line was used as a control line in these studies and cultured under the same conditions as iPSC lines.

All three iPSCs lines were cultured on irradiated mouse embryo fibroblasts using standard hESC media, containing DMEM/F12 (11330032, Gibco) supplemented with 20% KnockOut serum replacement (10828028, Gibco), 1x MEM Non-essential amino acids (11140050, Gibco), 1 mM L-glutamine (25030081, Gibco), 0.14% β-mercaptoethanol, and 8 ng/ml human FGF-basic recombinant protein (PHG0023, Gibco). Cells were kept in a humidified incubator set to 5% $CO_2$ at 37 °C. The medium was refreshed daily, and cells were passaged by cutting and pasting colonies every 6–7 days.

Differentiation was carried out using a dual-SMAD inhibition-based method using Noggin as the dual-SMAD inhibitor. Specifically, starting on day 0, iPSC colonies were maintained for 10 days in a N2B27 neural induction medium containing Neurobasal media (21103049,

Gibco) supplemented with 1% N-2 (17502048, Gibco), 1x B27 (17504044, Gibco), 2 mM L-glutamine (25030081, Gibco), 1% insulin-transferrin-selenium (51300044, Gibco), 500 ng/µl Noggin (6057-NG, R&D Systems), and 2 µM SB431542 (04-0010-05, Stemgent). Media was replaced on days 3, 5, 7, and 10. Subsequent feeds on days 11 and 13 used the N2B27 neural induction medium without Noggin and SB431542. By day 14, neuronal rosettes were carefully segmented into smaller clusters at a 1:2 ratio and seeded onto 6-well cell culture plate coated with poly-D-lysine (PDL, P0899, MilliporeSigma) and laminin (23017015, Gibco), replacing half the media every other day until the neural progenitor cells (NPCs) reached sufficient density. After approximately three weeks of proliferation, cells were dissociated using Accutase (A6964, MilliporeSigma), counted with a hemocytometer, and plated on PDL/laminin-coated cell culture plates in the above described N2B27 media supplemented with 10µM ROCK inhibitor Y-27632 (252-00701, Wako Chemicals) and replacing half the media every other day. For experimental use, neurons were dissociated using Accutase and plated on PDL/laminin-coated plates and terminally differentiated using Neuronal Differentiation medium comprising Neurobasal media supplemented with 1x B27, 1x MEM Non-essential amino acids, 2 mM L-glutamine, 10 ng/ml brain-derived neurotrophic factor (BDNF) (450-02, Peprotech), 10 ng/ml glial-derived neurotrophic factor (GDNF) (450-10, Peprotech), 200 µM ascorbic acid (A4544, MilliporeSigma), and 1 µM cyclic adenosine monophosphate (cAMP) (A9501, MilliporeSigma). To enhance cell attachment, 10 µM ROCK inhibitor was added during the initial plating. Half the media was replaced twice per week. After plating neuronal progenitor cells for terminal differentiation in Neuronal Differentiation medium, the H9 hESC and ASdel1-0 iPSC cultures were matured for 10 weeks, and ASdel1-0ΔI-P culture for 6.5 weeks.

### Immunofluorescence, high-content imaging, and dose-response analyses in mouse neurons

Immunofluorescence for high-content imaging analysis was performed as previously described with slight modifications[10]. Following drug treatment at DIV7, primary neurons were fixed at DIV10 with 4% paraformaldehyde in PBS at room temperature (RT) for 10 min. After three brief washes in PBS, neurons were permeabilized with PBS/1% Triton-X-100 at RT for 10 min, followed by blocking with PBS including 5% normal goat serum and 0.2% Triton-X-100 (NGST) at RT. For the screening, neurons were incubated with primary rabbit anti-GFP antibody (1:1000, Novus Cat# NB600-308, RRID:AB_10003058) at 4 °C overnight (ON). Neurons were then briefly washed with PBS followed by incubation with secondary goat anti-rabbit Alexa Fluor 488 antibody (1:500, Molecular Probes Cat# A-11008, RRID:AB_143165) and 4´,6-diamidino-2-phenylindole (DAPI, 1:7500, Thermo Fisher Scientific Cat# D1306, RRID:AB_2629482) at RT for 60 min. Following secondary antibody incubation, neurons were washed with PBS, and fluorescent images were acquired using a BD Pathway 855 bioimager (Becton Dickinson). All acquired images were analyzed by CellProfiler[48]. The mean GFP signal from each drug-treated well was normalized to the average mean GFP signal from DMSO-treated neurons. For the dose-response curves, an additional neuron-specific primary mouse anti-NeuN antibody (1:500, MilliporeSigma Cat# MAB377, RRID:AB_2298772) was used with a corresponding secondary anti-mouse Alexa Fluor 568 antibody (1:500, Thermo Fisher Scientific Cat# A-21124, RRID:AB_2535766). Fluorescent images were acquired using a Nikon Ti2 eclipse fluorescence microscope (Nikon Instruments Inc.) and analyzed using Nikon NIS-Elements software (Nikon Instruments Inc.). Antibody-enhanced UBE3A-YFP fluorescence intensities were determined from DAPI and NeuN double-positive neurons in drug-treated wells and normalized to neurons in DMSO-treated wells. The cytotoxicity of a compound was determined by calculating the percentage of neurons (DAPI and NeuN double-positive cells) alive in drug-treated wells compared to the DMSO-treated wells. For

determining the effectiveness of unsilencing, the percent neurons expressing GFP signal in neurons (DAPI and NeuN double-positive cells) in drug-treated wells was thresholded such that the percent GFP positive neurons in DMSO-treated wells was set to ~5%. Concentrations where more than 10% of neurons were dead compared to DMSO-treated wells were manually removed and the last included data point was set as a maximum efficacy. Sigmoidal curve fitting for the dose-response analysis was performed using GraphPad Prism software version 8.0 (GraphPad Software).

### Immunocytochemistry in human neurons

hESCs and iPSCs were differentiated for terminal differentiation on poly-D-lysine- and laminin-coated acid-treated coverslips. Once neurons reached 10 weeks of maturity, wells were treated with (S)-PHA533533 or 0.1% DMSO (vehicle) for 72 h. Neurons were fixed with 4% paraformaldehyde for 10 min at RT and then permeabilized using PBS plus 0.5% Triton X 100 (PBS-T) for 10 min at RT. Following permeabilization, neurons were blocked in 0.1% PBS-T containing 2% bovine serum albumin and 5% normal goat serum. Neurons were incubated with primary mouse anti-UBE3A (1:500, Sigma-Aldrich Cat# E8655, RRID:AB_261956) and rabbit anti-NeuN (1:300, Abcam Cat# ab177487, RRID:AB_2532109) antibodies in blocking buffer containing primary antibodies overnight at RT. The next day, the coverslips were washed three times 4 min with PBS followed by incubation with secondary goat anti-mouse Alexa Fluor 488 (1:400, Thermo Fisher Scientific Cat# A-11029, RRID:AB_2534088) and goat anti-rabbit Alexa Fluor 594 (1:400, Thermo Fisher Scientific Cat# A-11012, RRID:AB_2534079) for 3 h at RT in the dark. The coverslips were washed three times 4 min with PBS and mounted using ProLong™ Gold Antifade Mountant with DNA Stain DAPI (P36941, Invitrogen). Mounted coverslips were incubated for 24 h at RT in the dark before imaging. The images were captured using a 20X objective on a Zeiss Axio Observer Z1 microscope with consistent exposure times across all samples. Representative images were selected, and adjustments were made using ImageJ software. Only color balance was modified in the images, and this adjustment was applied uniformly to all samples.

### ASO-mediated knockdown and western blot analysis in mouse neurons

2' MOE gapmer antisense oligonucleotides (ASOs) designed to target mouse *Cdk2* sequence TTCTCCAAACGGGCCCTTGA, *Cdk5* sequence TAGGCTTACCTCCTACCAAA, *Top1* sequence AACAGCTCGATTGGCACGGT, and scrambled GCGACTATACGCGCAATATG sequence as a negative control (Integrated DNA Technologies) were formulated in water and added to cultured cortical neurons grown on 6-well plates at DIV5 at a final concentration of 10 µM. Neurons were further cultured and treated at DIV7 with vehicle (0.1% DMSO) or the indicated compounds for 72 h. For western blot analysis, neurons were washed once with PBS and then lysed using RIPA buffer (50 mM Tris-HCl pH 8.0, 150 mM NaCl, 1% NP−40, 0.5% Na-deoxycholate, 0.5% SDS) and protease inhibitor cocktail (P8340, MilliporeSigma) on ice. The extracts were sonicated and centrifuged at 4 °C to remove insoluble cell debris. Protein concentrations were determined by using BCA protein assay kit (Pierce). A total of 30 µg of each sample was separated in 7.5% Mini-PROTEAN TGX precast protein gel (Bio-Rad) by electrophoresis and transferred onto the 0.45 µm Immobilon-FL PVDF membrane (MilliporeSigma) in ice-cold transfer buffer (25 mM Tris-base, 192 mM glycine, and 20% MeOH) at 90 mA for 90 minutes. Membranes were blocked in Odyssey Blocking Buffer for 1 hour and then blotted with primary antibodies [anti-UBE3A (1:1000, Sigma-Aldrich Cat# SAB1404508, RRID:AB_10740376), anti-GFP (1:1000, Novus Cat# NB600-308, RRID:AB_10003058), anti-CDK5 (1:500, Santa Cruz Biotechnology Cat# sc-6247, RRID:AB_627241), anti-CDK2 (1:500, Abcam Cat# ab64669, RRID:AB_1603771), anti-TOP1 (1:1000, Santa Cruz Biotechnology Cat# sc-271285, RRID:AB_10611597), anti-GAPDH (1:1000,

**Table 2 | Sequences of mouse primers used for RT-qPCR**

| Primer name | Sequence |
| --- | --- |
| Ube3a_Fwd | CAAAAGGTGCATCTAACAACTCA |
| Ube3a_Rev | GGGGAATAATCCTCACTCTCTC |
| YFP_Fwd | ACATGAAGCAGCACGACTTCT |
| YFP_Rev | GACGTTGTGGCTGTTGTAGTTGTA |
| Gapdh_Fwd | GGTCGGTGTGAACGGATTT |
| Gapdh_Rev | GATGGCAACAATCTCCACTTTG |
| Snord115_Fwd | CTGGGTCAATGATGACAAC |
| Snord115_Rev | TTGGGCCTCAGCGTAATCC |
| Snord116_Fwd | GGATCTATGATGATTCCCAG |
| Snord116_Rev | GGACCTCAGTTCCGATGA |
| Ube3a-ATS_Fwd | ACAGAACAATAGGTCACCAGGTT |
| Ube3a_ATS_Rev | AAGCAAGACTGTTCACCTCAT |
| Snrpn_Fwd | TTGGTTCTGAGGAGTGATTTGC |
| Snrpn_Rev | CCTTGAATTCCACCACCTTG |
| Dlg2_Fwd | GACCGGGCGATTAAACTAGAA |
| Dlg2_Rev | GGTCCAGACTGCTCTTCAATAA |
| Nrxn3_Fwd | TGCCACCTGAAATGTCTACC |
| Nrxn3_Rev | ATCTGACGTGGGCTGAATG |
| Malat1_Fwd | GCGGTGTCTTTGCTTGACTC |
| Malat1_Rev | AGGTGTGTCGACTCAGAGGT |
| Pantr1_Fwd | GCCCGGGACTGTAAGGCGGATA |
| Pantr1_Rev | GGCACTCTGTCCCTCTCCCTCG |
| Airn_Fwd | CGTGCTCTCCACAGCTCTTGCC |
| Airn_Rev | CGGCTTCTTGCAGCTTCTGAGG |
| UBE3A_Fwd | AGTCTGACGACATTGAAGCTAGCC |
| UBE3A_Rev | AAGGTAAGCTGAGCTTGCTCC |
| GAPDH_Fwd | GCTTAGCACCCCTGGCCAAGG |
| GAPDH_Rev | CTTGGCAGCGCCAGTAGAGG |

MilliporeSigma Cat# MAB374, RRID:AB_2107445), anti-β-Tubulin (1:10,000, Abcam Cat# ab6046, RRID:AB_2210370), anti-β-Actin (1:5000, MilliporeSigma Cat#A1978, RRID:AB_476692)] overnight at 4 °C. The next day, the membranes were washed with PBS/0.5% Tween-20 three times and incubated with anti-mouse or anti-rabbit HRP-conjugated secondary antibodies (1:5000, Thermo Fisher Scientific Cat# 31430, RRID:AB_228307 or Thermo Fisher Scientific Cat# 31460, RRID:AB_228341, respectively) for 1 h at RT. Chemiluminescence reaction was performed using Clarity Western ECL Substrate (Bio-Rad), which was imaged by an Amersham Imager 680 (GE Healthcare). The signal was quantified using the ImageJ software (NIH).

## RNA isolation and RT-qPCR of mouse samples
For RNA isolation, cultured cortical neurons were lysed using 350 μl of RLT lysis buffer (Qiagen) per well on a 6-well plate. For isolating RNA from mouse brain tissue, microdissected brain regions were snap-frozen in liquid nitrogen and stored at −80 °C until the day of RNA extraction, and then lysed using RLT lysis buffer containing 1% of β-mercaptoethanol. Total RNA was extracted from both cultured cells and brain tissue using RNeasy Mini kit (Qiagen). cDNA was synthesized from 1 μg of total RNA using qScript cDNA Supermix (Quantabio). 1/25 of synthesized cDNA was used per qPCR reaction together with PowerUp SYBR Green Master Mix (A25742, Applied Biosystems) and a specific primer pair (Table 2). qPCR was performed in a QuantStudio 5 Real-Time PCR system (Applied Biosystems). The specificity of the amplification products was verified by melting curve analysis. All qPCRs were conducted in technical triplicates, and the results were averaged for each sample, normalized to *Gapdh* expression, and analyzed using the comparative CT method (ΔΔCT).

## RNA isolation and RT-qPCR of human samples
Total RNA was isolated from harvested neurons using RNA-Bee (AMS Biotechnology) or RNA Stat-60 (AMS Biotechnology) according to the manufacturer's protocol. cDNA was produced using the High Capacity cDNA Reverse Transcription Kit (Life Technologies). RT-qPCR was performed in triplicate using TaqMan Gene Expression Assays (Life Technologies) according to the manufacturer's recommendations. The following assays were used: *UBE3A-ATS* (SNRPN-Hs01372960_m1), *UBE3A* (UBE3A-Hs00166580_m1), and *GAPDH* (GAPDH-Hs99999905_m1).

## Western blot analysis of human samples
hESCs and iPSCs were plated for terminal differentiation in a 12-well plate at a density of ~250,000 cells/well. Once neurons reached 10 weeks of maturity, wells were treated with (*S*)-PHA533533 or 0.1% DMSO (vehicle) for 72 h. Neurons were harvested by scraping into ice-cold PBS, pelleted via centrifugation at 4 °C, and then flash-frozen in liquid nitrogen before storage at −80 °C. Two wells of each treatment were collected and combined upon thawing in 100 μl of cell lysis buffer (#9803, Cell Signaling Technology) containing protease inhibitors [1 mM phenylmethylsulfonyl fluoride (PMSF) and 1x Protease Inhibitor Cocktail Set III (#539134, Calbiochem)]. Protein concentrations were determined by BCA assay and 22 μg of protein in Laemmli Sample Buffer (#1610747, Bio-Rad) was separated by SDS-PAGE using 4–20% Mini-PROTEAN® TGX™ Precast Protein Gel (#4568093, Bio-Rad). Protein was transferred to PVDF membrane using the TransBlot Turbo system (Bio-Rad). The membrane was blocked using 5% w/v non-fat milk in 1X TBS (#1706435, Bio-Rad) plus 0.05% Tween-20 (TBS-T) at room temperature (RT) for 1 h. The membrane was first incubated with primary mouse anti-UBE3A (1:1000; Sigma-Aldrich Cat# E8655, RRID:AB_261956) antibody in blocking buffer overnight at 4 °C. The next day, the membranes were washed with TBS-T three times for 10 min and incubated with anti-mouse HRP-conjugated secondary antibody (1:2000, Cell Signaling Technology Cat# 7076, RRID:AB_330924) in blocking buffer for 1 h at RT, following another three times 10 min washes. Chemiluminescence reaction was performed using Clarity Western ECL Substrate (#1705060, Bio-Rad) and imaged on the ChemiDoc Touch imaging system (Bio-Rad). Following the imaging of UBE3A, the membrane was then incubated with primary mouse anti-GAPDH (1:10,000, Sigma Cat# MAB374, RRID:AB_2107445) antibody in blocking buffer for 1 h at RT for loading control. The signal was quantified using the ImageJ software (NIH).

## In vivo drug treatment and immunohistochemistry
Either (*S*)-PHA533533 (2 mg/kg in 10 ml/kg body weight) formulated in saline (0.9% NaCl) or 0.9% saline (vehicle control) was injected i.p. into postnatal day 11 (P11) *Ube3a*^{m−/p+} or WT mice. At P13, mice were deeply anesthetized with euthasol (100 mg/kg, i.p.) prior to transcardial perfusion with PBS, pH 7.4, followed by phosphate-buffered 4% paraformaldehyde, pH 7.4. Perfused brains were postfixed overnight and cryoprotected with 30% sucrose in PBS, pH 7.4, for 2 days. Cryoprotected brains were frozen on dry ice and cut into 40 μm-thick sections with an Epredia HM 430 Sliding Microtome (Thermo Scientific). Sections were stored in a cryopreservative solution (by volume: 45% PBS, 30% ethylene glycol, 25% glycerol) at −20 °C until they were processed for free-floating immunohistochemistry. For immunohistochemistry, sections were washed three times with PBS, followed by a 15-min treatment with 1% sodium borohydride in PBS to reduce nonspecific immunoreactivity. Sections were washed thrice with PBS before blocking with 1% BSA/PBS/0.1% Triton X-100 for 30 min. Sections were incubated with mouse anti-UBE3A (1:1000, Sigma-Aldrich Cat# SAB1404508, RRID:AB_10740376) and guinea pig anti-NeuN (1:1000, MilliporeSigma Cat# ABN90, RRID:AB_11205592) primary antibodies diluted in PBS/0.1% Triton X-100 (PBST) for 24 h at 4 °C. Sections were

then washed three times with PBST and blocked again before incubating with goat anti-mouse Alexa Fluor 488 (1:500, Thermo Fisher Scientific Cat# A-21131, RRID:AB_2535771) and anti-guinea pig Cy3 (1:500, Jackson ImmunoResearch Labs Cat# 706-165-148, RRID:AB_2340460) secondary antibodies together with DAPI (1:7500, D1306, Invitrogen) all diluted in PBST for 24 h at 4 °C. Sections were then washed three times with PBST, followed by three washes with PBS before mounting the sections on gelatin-coated slides using Vectashield Vibrance Antifade Mounting Medium (Vector Laboratories Inc.). Images were acquired using a Stellaris 8 Falcon confocal microscope (Leica Microsystems). Images compared within the same figures were taken using identical imaging parameters from sections stained within the same experiment.

### Western blot and RT-PCR in HEK293T cells

HEK293T cells (CRL-3216, ATCC) cultured in DMEM (11995065, Gibco) supplemented with 10% FBS (F4135, MilliporeSigma) were plated on a 12-well plate at a density of 400,000 cells per well. Six hours later, cells were treated with either 0.1% DMSO (vehicle) or (S)-PHA533533 for 48 h. For western blot analysis, cells were washed once with PBS and then lysed using RIPA buffer (50 mM Tris-HCl pH 8.0, 150 mM NaCl, 1% NP-40, 0.5% Na-deoxycholate, 0.5% SDS) and protease inhibitor cocktail (P8340, MilliporeSigma) on ice. The extracts were sonicated and centrifuged at 4 °C to remove insoluble cell debris. Protein concentrations were determined by using BCA protein assay kit (Pierce). A total of 30 μg of each sample was separated in 7.5% Mini-PROTEAN TGX precast protein gel (Bio-Rad) by electrophoresis and transferred onto the 0.45 μm Immobilon-FL PVDF membrane (MilliporeSigma) in ice-cold transfer buffer (25 mM Tris-base, 192 mM glycine, and 20% MeOH) at 90 mA for 90 minutes. Membranes were blocked in Odyssey Blocking Buffer for 1 hour and then blotted with primary antibodies anti-UBE3A (1:1000, Sigma-Aldrich Cat# SAB1404508, RRID:AB_10740376) and anti-GAPDH (1:1000, MilliporeSigma Cat# MAB374, RRID:AB_2107445) overnight at 4 °C. The next day, the membranes were washed with PBS/0.5% Tween-20 three times 10 min and incubated with anti-mouse or HRP-conjugated secondary antibodies (1:5000, Thermo Fisher Scientific Cat# 31430, RRID:AB_228307, Invitrogen) for 1 h at RT. Chemiluminescence reaction was performed using Clarity Western ECL Substrate (Bio-Rad), which was imaged by an Amersham Imager 680 (GE Healthcare). The signal was quantified using the ImageJ software (NIH). For RT-PCR isolation, HEK293T cells were lysed using 250 μl of RLT lysis buffer (Qiagen) per well on a 12-well plate. Total RNA was extracted using RNeasy Mini kit (Qiagen). cDNA was synthesized from 1 μg of total RNA using qScript cDNA Supermix (Quantabio). 1/25 of synthesized cDNA was used per qPCR reaction together with PowerUp SYBR Green Master Mix (A25742, Applied Biosystems) and primers for *GAPDH* and *UBE3A* (Table 2). qPCR was performed in a QuantStudio 5 Real-Time PCR system (Applied Biosystems). The specificity of the amplification products was verified by melting curve analysis. All qPCRs were conducted in technical triplicates, and the results were averaged for each sample, normalized to *GAPDH* expression, and analyzed using the comparative CT method (ΔΔCT).

### Statistical analysis

Neuron cultures were obtained from different litters and prepared separately. For the in vivo experiments, each data point represents an independent animal. Data from the qPCR and Western blot in vitro assays were first log-transformed to obtain a normal distribution of the data and then autoscaled before statistical analysis according to Willems et al.[49]. All statistical testing was performed using GraphPad Prism software version 8.0 (GraphPad Software).

### Reporting summary

Further information on research design is available in the Nature Portfolio Reporting Summary linked to this article.

## Data availability

iPSC cell lines are available upon request and after completion of Material Transfer Agreements through the University of Connecticut Cell and Genome Engineering Core. Source data are provided with this paper and are available at https://doi.org/10.6084/m9.figshare.26005021.v1.

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

## Acknowledgements

This work was supported by the Angelman Syndrome Foundation, the NINDS (R01NS131615 to B.D.P. and J.A., and R01NS129914 to B.D.P.), the NICHD (R01HD094953 to S.J.C.), the Simons Foundation (SFARI grant #702556 to B.D.P.), a sponsored research agreement between the University of North Carolina and Pfizer (to B.D.P.), and Pinnacle Hill, LLC, a portfolio company of certain funds managed by Deerfield Management Company, L.P (to B.D.P.). We would like to thank the Rare Disease Research Unit at Pfizer for supporting the primary screening of the Pfizer chemogenetic library. We thank the Compound Transfer Program of Pfizer for additionally providing PHA533533 and its analogs. Microscopy was performed at the UNC Neuroscience Microscopy Core (RRID:SCR_019060), supported, in part, by funding from the NIH-NINDS Neuroscience Center Support Grant P30 NS045892 and the NIH-NICHD Intellectual and Developmental Disabilities Research Center Support Grant P50 HD103573. We are grateful to Dr. Matthew Judson from the University of North Carolina, Chapel Hill for discussions and critical reading of the manuscript.

## Author contributions

H.V., A.W.A., H.M.L., J.L.Cotney, H.C.H., J.L.Collins, S.J.C., J.A., and B.D.P. contributed intellectually to the experimental design. H.V., H.M.L., and K.R.B. performed the drug screen and initial validations. A.W.A. and K.L. synthesized and purified (*S*)- and (*R*)-PHA533533. H.V., K.R.B., A.S., and E.G. performed pharmacokinetic analyses of the analogs. H.V., A.S., and E.G. performed all mouse RT-qPCR and western blot assays. H.V., K.R.B., and A.S. performed in vivo studies and immunofluorescence. R.B.G. performed all human iPSC experiments. H.V. analyzed data and wrote the manuscript. All authors reviewed and edited the manuscript.

## Competing interests

The authors performed the initial drug screen, shown in Fig. 1b, as part of a sponsored research agreement with Pfizer that (1) provided the chemogenetic library, (2) partially supported H.-M.L.'s salary, and (3) provided research funds to perform the initial screen in the laboratory of B.D.P. No other parts of the project were supported by Pfizer. Pfizer reviewed and approved the manuscript submission, but did not contribute to the conceptualization, design, data collection, data analysis, or preparation of the manuscript. The authors do not have any other competing interests.
