## [Peer Review File · Nature Communications]

Ube3a unsilencer for the potential treatment of Angel man syndromeREVIEWER COMMENTS

Reviewer #1 (Remarks to the Author):

The submitted work reveals a novel small molecular unsilencer of the paternal Ube3A gene, S-PHA533533, in both mouse and human neurons. The authors have rigorously characterized the potency and mechanism of S-PHA533533 unsilencing of Ube3A and demonstrated that this is due to inhibition of a Ube3A antisense transcript from the paternal allele and it not due to inhibition of topoisomerase 1 activity, as was the case for another drug, topotecan, previously discovered by this group. The authors also demonstrate that S-PHA533533 is not unsilencing through known mechanisms of Cdk2 or Cdk5 inhibition. The authors also performed a chemical optimization of the compound and identified specific chemical properties of S-PHA533533 that promote unsilencing. Interestingly, the R-enantiomer of PHA533533 is ineffective. The authors demonstrate that S-PHA533533 is effective when injected systemically in vivo in mice to increase Ube3A levels throughout the brain as well as in human iPSC derived neurons from an Angelman Syndrome patient. These results provide strong evidence for the potential for S-PHA533533 as a lead compound and novel treatment strategy for Angelman Syndrome.

I only have a couple of comments:

1. Does the S-PHA533533 affect expression of other known antisense transcripts in neurons or is it specific for the Ube3A ATS? This may provide insight into its specificity and mode of action.
2. The authors discuss at length (primarily in the discussion) the pitfalls of other methods to unsilenced the Ube3A allele or decrease Ube3A ATS. A discussion that is more focused on the results in the manuscript would be better.

Reviewer #2 (Remarks to the Author):

This manuscripts describes the identification and proof-of-concept characterisation of a novel "unsilencer" for the treatment of Angelman syndrome by reactivation of the normally silenced paternal copy of imprinted UBE3A. The manuscript is really clearly written and describes a series of in vitro and in vivo experiments that lay the ground-work for further development of this potential therapy. Although they do not completely delineate the mode-of-action of this drug, the authors point out this is not necessarily required for small molecules to advance, and I also do not think it is an issue that should preclude publication. Overall the manuscript is excellent and I do not have any major concerns. I do have three minor queries that need clarification, detailed below.

1. When testing whether (S)-PHA533533 treatment might impair transcriptional elongation and reduce the expression of other long genes in addition to Ube3a-ATS (as topotecan does), the authors examine expression of Dlg2 and Nr3n3 in primary neuron cultures (Fig. 4). I wonder why they have focused on these, admittedly, long genes rather than other lncRNA species? (Possibly even more insightful would be to look at other imprinted lncRNAs...)
2. The authors choose a much lower dose (2mg/Kg) to treat their mouse model than those previously published (7.5mg/Kg). I wonder how they came about this dosing level?
3. In the final set of experiments the authors use iPSC models to investigate the effectiveness of (S)-PHA533533 treatment in a human models system. This may be displaying my misunderstanding but why (in Figure 6B) do the DMSO treated samples have any UBE3A expression? My understanding was that the deletion of the boundary element in these cells restricts imprinting of UBE3A to neurons (thus allowing differentiation) - so why then is there any UBE3A expression in this model of AS? Although (S)-PHA533533 treatment doubles UBE3A expression, it isn't clear what that is relative to? For instance, does it restore UBE3A expression to normal levels? Also, how would this compare to topotecan?

I choose to waive my anonymity - Anthony Isles

Reviewer #3 (Remarks to the Author):

The authors have found a new compound that reactivates UBE3A protein expression from the paternal allele through downregulation of the ube3a-ATS. (S)-PHA533533 (PHA5) works in this manner both in mutant mouse models for maternal loss of function UBE3A and in patient derived neurons. Given that loss of maternal ube3a is associated with the neurodevelopmental disorder Angelman syndrome, PHA5 serves as a potential therapeutic for treatment. While previous similar efforts targeting the ube3a-ATS have had difficulties, the authors demonstrate that PHA5 overcomes several of these hurdles through non-invasive delivery and brain-wide activation of the paternal UBE3A. Given the importance of UBE3A to human brain function in health and disease, this work is majorly significant. Also, it provides a tool compound for others to use in their own research related to many questions about UBE3A.

Minor comments:

Figure 1:

- a) Please point to a baseline (DMSO) signal in figure 1B plot.
- b) Most compounds strongly suppress UBE3A-YFP signal. Given that these mice should not express any UBE3A-YFP, why would the authors see significant decrease in signal? Is the background coming from non-neuronal cells? It would have been interesting to look at several of these strong inhibitor compounds. Maybe they are just killing the cell. The authors should say something about it.
- c) I am surprised that the authors have found such a great compound in the 2800 compound library. In the future, using this great mouse line it may be worth considering the FDA approved libraries or the much larger billion compound libraries. In the absence of this, the authors should discuss the reason for using this library. What is its complexity? What types of compounds are there in the library that increases confidence of broad survey of compound types with unique features?
- d) Please indicate time of treatment for figure 1b.
- e) Was Figure 1b accomplished through live imaging or imaging of fixed and stained cells? If fixed and stained, include these details in your figure 1a.

Figure 2:

- a) Please include some stats on these graphs?
- b) I was not clear on what the authors mean by "Viability (% of DMSO)". Why do the cells die with more paternal UBE3A? Make the discussion of this figure as clear as possible.
- c) PHA5 should be tested in a non-neuronal cell? What happens to ube3a levels in the non-neuronal cells that express the paternal allele and are treated with PHA5?
- d) Show total levels of UBE3A in wild type neurons treated with PHA5.
- e) What is the time course of treatment for these assays?
- f) Counting NeuN and Dapi, I do not believe is definitive for viability measurements. It is possible that neurons are detaching from the plate and yet still alive. Maybe change the title to percent of cells. One thought is whether the decrease in cell density is giving rise to the increase in % neurons that have UBE3A-YFP.

Figure 3:

- a) Put molecular weight markers on each immunoblot. I was having trouble telling whether I am looking at endogenous UBE3A or UBE3A-YFP. Is this full length UBE3A?
- b) What is the mRNA level of ube3a in WT neurons treated with PHA5?
- c) What is the mRNA level of ube3a in a non-neuronal cell treated with PHA5?

Figure 4:

- a) Put molecular weight markers on each immunoblot. I was having trouble telling whether I am looking at endogenous UBE3A or UBE3A-YFP. Is this full length UBE3A?
- b) What happens to total non-YFP labelled maternal ube3a in the PHA5 treated neurons in this figure? Please show.

Figure 5:

- a) I was having trouble understanding the data in 5A compared to 5B-5D. The differences by fluorescence appear to be far greater than what is quantified in 5A. The fluorescence does not appear to be a 50% difference. It appears to be a far greater difference. This may be a small point; however, the authors should discuss it and make every effort to explain why they may see these differences compared to their immunoblots and mRNA measurements.

Figure 6:

- a) What did these neurons look like? The authors should show other neuronal markers and how they change with these treatments.
- b) Would be great to see an immunoblot of these treated samples?
- c) Did the UBE3A protein level go up?
- d) How about viability measurements in these cells?

Overall:

This is an interesting study. A model of how this compound may be working would be a nice addition to the study. The major benefit of this study is the identification of the compound. Using the compound in future studies and sharing with AS community may have major positive implications for AS research and possibly the patient population.

Author response to reviewers:

We thank the reviewers for their comments and positive feedback, which has helped us to strengthen the manuscript. Please find our detailed point-by-point response to all reviewer comments in blue below.

We have added new experiments in response to reviewer concerns, including:

- Quantitative RT-PCR experiments to measure whether (S)-PHA533533 or topotecan treatment alters other lncRNAs, including those acting on antisense transcripts and imprinted genes, in mouse primary neurons (new **Figure 4k-m**).
- Quantitative RT-PCR and western blot analysis to demonstrate that both (S)-PHA533533 and topotecan increase *Ube3a* mRNA and protein levels in WT mouse primary neurons (new **Supplementary Figure 2**).
- Western blot analysis and immunocytochemistry to demonstrate the increase in UBE3A protein levels in pluripotent stem cell-derived neurons from an individual with AS following (S)-PHA533533 treatment (new **Supplementary Figure 3**).
- Quantitative RT-PCR and western blot analysis to demonstrate that (S)-PHA533533 does not change the mRNA and protein levels of UBE3A in non-neuronal cells, such as HEK293T cells, that express *UBE3A* biallelically (new **Supplementary Figure 4**).

In addition to addressing all reviewer criticisms, we have made some additional minor editorial corrections to our previous version of the manuscript, the most notable being correcting an axis label in a figure:

- Corrected the label of the x-axes in **Figure 3f-g** from “Norm. patUbe3a mRNA levels” and “Norm. patUBE3A protein levels” to “Norm. Ube3a mRNA levels” and “Norm. UBE3A protein levels”, respectively.

Reviewer #1 (Remarks to the Author):

The submitted work reveals a novel small molecular unsilencer of the paternal *Ube3A* gene, S-PHA533533, in both mouse and human neurons. The authors have rigorously characterized the potency and mechanism of S-PHA533533 unsilencing of *Ube3A* and demonstrated that this is due to inhibition of a *Ube3A* antisense transcript from the paternal allele and it not due to inhibition of topoisomerase 1 activity, as was the case for another drug, topotecan, previously discovered by this group. The authors also demonstrate that S-PHA533533 is not unsilencing through known mechanisms of Cdk2 or Cdk5 inhibition. The authors also performed a chemical optimization of the compound and identified specific chemical properties of S-PHA533533 that promote unsilencing. Interestingly, the R-enantiomer of PHA533533 is ineffective. The authors demonstrate that S-PHA533533 is effective when injected systemically in vivo in mice to increase *Ube3A* levels throughout the brain as well as in human iPSC derived neurons from an Angelman Syndrome patient. These results provide strong evidence for the potential for S-PHA533533 as a lead compound and novel treatment strategy for Angelman Syndrome.

I only have a couple of comments:

1. Does the S-PHA533533 affect expression of other known antisense transcripts in neurons or is it specific for the *Ube3A* ATS? This may provide insight into its specificity and mode of action.

We performed additional experiments to understand whether (S)-PHA533533 could affect the expression of long non-coding RNAs (lncRNAs), including those acting as antisense transcripts, in neurons. To answer this, we measured the expression levels of *Malat1*, *Pantr1*, and *Airn* upon (S)-PHA533533 and topotecan treatment, compared to vehicle controls, in primary neurons. Our new

results, depicted in **Figure 4k-m**, show that only topotecan treatment reduces the expression of *Malat1*, which is one of the longest lncRNAs (~9kb). Neither topotecan nor (S)-PHA533533 treatment affected the expression of *Pantr1*, an intergenic antisense lncRNA that shares a bidirectional promoter with *Pou3f3* gene, or *Airn*, a paternally expressed antisense lncRNA to *Igf2r1*. Based on our results, we do not see evidence that (S)-PHA533533 has a general effect on the expression of antisense transcripts. While we did not observe any effects in our candidate lncRNAs, we appreciate that there may be other lncRNAs that are affected, as it is impractical to examine them all. A future effort will be to perform a more exhaustive analysis of lncRNAs, but the new data indicate there is no general effect of (S)-PHA533533 treatment on lncRNAs or antisense transcripts.

We now provide these new details in our Results, Discussion, and **Figure 4k-m**.

2. The authors discuss at length (primarily in the discussion) the pitfalls of other methods to unsilenced the Ube3A allele or decrease Ube3A ATS. A discussion that is more focused on the results in the manuscript would be better.

We have revised and shortened the discussion section to prioritize the presentation of our results and their implications, minimizing the discussion of other methods. While we believe some discussion of other methods is warranted to contextualize the importance of our findings, we attempted to provide a balanced view that acknowledges that all therapeutic approaches have strengths and weaknesses, and we have provided a discussion of the limitations and future directions of our current approach. We believe that the re-focused discussion enhances the clarity and focus of the manuscript, and addresses the reviewer's concerns by better emphasizing the significance of our findings.

Reviewer #2 - (Remarks to the Author):

This manuscript describes the identification and proof-of-concept characterisation of a novel "unsilencer" for the treatment of Angelman syndrome by reactivation of the normally silenced paternal copy of imprinted UBE3A. The manuscript is really clearly written and describes a series of in vitro and in vivo experiments that lay the ground-work for further development of this potential therapy. Although they do not completely delineate the mode-of-action of this drug, the authors point out this is not necessarily required for small molecules to advance, and I also do not think it is an issue that should preclude publication. Overall the manuscript is excellent and I do not have any major concerns. I do have three minor queries that need clarification, detailed below.

1. When testing whether (S)-PHA533533 treatment might impair transcriptional elongation and reduce the expression of other long genes in addition to Ube3a-ATS (as topotecan does), the authors examine expression of *Dlg2* and *Nrxn3* in primary neuron cultures (Fig. 4). I wonder why they have focused on these, admittedly, long genes rather than other lncRNA species? (Possibly even more insightful would to look at other imprinted lncRNAs...)

Reviewer #1 asked a similar question, and we repeat our response here for convenience:

We performed additional experiments to understand whether (S)-PHA533533 could affect the expression of long non-coding RNAs (lncRNAs), including those acting as antisense transcripts, in neurons. To answer this, we measured the expression levels of *Malat1*, *Pantr1*, and *Airn* upon (S)-PHA533533 and topotecan treatment, compared to vehicle controls, in primary neurons. Our new results, depicted in **Figure 4k-m**, show that only topotecan treatment reduces the expression of *Malat1*, which is one of the longest lncRNAs (~9kb). Neither topotecan nor (S)-PHA533533 treatment affected the expression of *Pantr1*, an intergenic antisense lncRNA that shares a bidirectional

promoter with *Pou3f3* gene, or *Airn*, a paternally expressed antisense lncRNA to *Igf2r*¹. Based on our results, we do not see evidence that (S)-PHA533533 has a general effect on the expression of antisense transcripts. While we did not observe any effects in our candidate lncRNAs, we appreciate that there may be other lncRNAs that are affected, as it is impractical to examine them all. A future effort will be to perform a more exhaustive analysis of lncRNAs, but the new data indicate that there is no general effect of (S)-PHA533533 treatment on lncRNAs or antisense transcripts.

We now provide these new details in our Results, Discussion, and **Figure 4k-m**.

2. The authors choose a much lower dose (2mg/Kg) to treat their mouse model than those previously published (7.5mg/Kg). I wonder how they came about this dosing level?

We appreciate the reviewer's inquiry regarding our choice of a lower dose (2 mg/kg) for (S)-PHA533533 compared to the previously published dose of 7.5 mg/kg. The higher dose of 7.5 mg/kg was utilized in studies involving adult mice, whereas our study specifically targeted younger (P11) mice, an age that falls within the optimal postnatal treatment window for AS mice.

Recognizing the potential differences in drug metabolism, tolerance, and developmental vulnerabilities between adult and young mice, we conducted a preliminary dose-response evaluation to determine the optimal dosage for our P11 treatment model. Through systematic testing of different doses, we found that the 2 mg/kg dosage was well-tolerated and sufficient to produce unsilencing, whereas we observed evidence of toxicity at 5 mg/kg in these young mice. We hypothesize that the lower tolerance threshold observed at this developmental stage could be attributed to the ongoing critical developmental processes, as well as the potent inhibitory effects of the drug on CDK2/CDK5. We believe that our choice of dose reflects a thoughtful consideration of the unique developmental context of our study and ensures the safety and efficacy of the treatment regimen for the specific age group under investigation. In the future, it will be essential to conduct large-scale juvenile animal toxicology studies to ascertain the safety profile of our novel unsilencer across development.

We have provided further clarification on this rationale in the Results section.

3. In the final set of experiments the authors use an iPSC models to investigate the effectiveness of (S)-PHA533533 treatment in a human models system. This may be displaying my misunderstanding but why (in Figure 6B) do the DMSO treated samples have any UBE3A expression? My understanding was that the deletion of the boundary element in these cells restricts imprinting of UBE3A to neurons (thus allowing differentiation) - so why then is there any UBE3A expression in this model of AS?

Briefly, not all the cells in the human iPSC model have differentiated into mature neurons, and some glia also exist in the cultures. Because both immature neurons and glia biallelically express *UBE3A*, there is always a background level of *UBE3A* in these preparations.

To expand on this, *UBE3A* imprinting is influenced not only by the presence or absence of the boundary element, but also by the maturity of neuronal cultures. As demonstrated in the original paper describing the ASdel1-0ΔI-P iPSC, the paternal silencing of *UBE3A* requires both the removal of the boundary element and sufficient expression of *UBE3A-ATS/SNRPN*² (Figure 4 in the Hsiao *et al.*² paper). However, the paternal silencing of *UBE3A* in these cultures can be influenced by various factors, including the heterogeneity of cell types produced during differentiation and the relative maturity of the cells, which can be affected by plated cell density.

We have now emphasized these points in the manuscript.

Although (S)-PHA533533 treatment doubles UBE3A expression, it isn't clear what that is relative to? For instance, does it restore UBE3A expression to normal levels? Also, how would this compare to topotecan?

In **Figure 6**, the doubling of *UBE3A* mRNA expression following (S)-PHA533533 treatment in human neurons differentiated from ASdel1-0 Δ I-P iPSCs is relative to the vehicle-treated condition. Unfortunately, due to the size of the deletion the AS patient harbors, there is no feasible way to create an isogenic control for that cell line with available genome editing technologies for comparison. However, we did perform experiments to assess UBE3A levels against a “wildtype”, non-isogenic, H9 cell line. These experiments suggest that (S)-PHA533533 treatment increases UBE3A protein levels in 10-week-old induced pluripotent stem cell-derived neurons from an individual with Angelman syndrome (ASdel1-0), reaching about 90% of UBE3A expressed in human neurons differentiated from H9 hESC control line cultured under the same conditions (new **Supplementary Figure 3**). However, we readily recognize that any comparison between these cell lines (H9 vs ASdel1-0) can be phenomenological at best, and cannot be considered a proper control, thus we think this experiment is best kept as supplementary data.

Although we did not use topotecan as a control unsilencer in our experiments in human neurons, Sen *et al.*³ have demonstrated that treating 11-week human cortical organoids derived from ASdel1-0 iPSCs for 72 hours with 1 μ M topotecan increases *UBE3A* mRNA levels by 1.8-fold (Figure 4 in the Sen *et al.*³ paper), suggesting comparative unsilencing capabilities of (S)-PHA533533 and topotecan in human neurons.

Reviewer #3 (Remarks to the Author):

The authors have found a new compound that reactivates UBE3A protein expression from the paternal allele through downregulation of the ube3a-ATS. (S)-PHA533533 (PHA5) works in this manner both in mutant mouse models for maternal loss of function UBE3A and in patient derived neurons. Given that loss of maternal ube3a is associated with the neurodevelopmental disorder Angelman syndrome, PHA5 serves as a potential therapeutic for treatment. While previous similar efforts targeting the ube3a-ATS have had difficulties, the authors demonstrate that PHA5 overcomes several of these hurdles through non-invasive delivery and brain-wide activation of the paternal UBE3A. Given the importance of UBE3A to human brain function in health and disease, this work is majorly significant. Also, it provides a tool compound for others to use in their own research related to many questions about UBE3A.

Minor comments:

Figure 1:

a) Please point to a baseline (DMSO) signal in figure 1B plot.

Thank you for pointing out that the color of the baseline (DMSO) was not noticeable. We have changed the color of the baseline (DMSO) from gray to green in **Figure 1b**, which improves visualization.

b) Most compounds strongly suppress UBE3A-YFP signal. Given that these mice should not express any UBE3A-YFP, why would the authors see significant decrease in signal? Is the background coming from non-neuronal cells? It would have been interesting to look at several of these strong inhibitor compounds. Maybe they are just killing the cell. The authors should say something about it.

To identify novel unsilencers of paternal *Ube3a*, we employed a previously developed high-content small molecule screen in cultured mouse primary neurons derived from paternal *Ube3a-YFP* mice⁴. However, these cultures do contain a small population of glial cells that have biallelic *Ube3a* expression and therefore express UBE3A-YFP protein, as also evident by immunostaining of the DMSO-treated culture in **Figure 1c**. To clarify this, we have added a sentence explaining why we have a background UBE3A-YFP signal in our neuronal cultures under the paragraph describing the results in **Figure 1** and **Figure 3**. We have also stated: “*Note that the low but detectable Ube3a levels in DMSO and (R)-PHA533533-treated Ube3a^{m-/p+} cells likely arise from the small population of glial cells and immature neurons, which biallelically express Ube3a, present in our neuronal cultures*”. Because our neuronal cultures contain a small population of glial cells and immature neurons expressing paternal *Ube3a-YFP*, the decrease in UBE3A-YFP signal could come from decreased paternal UBE3A levels in glial cells, decreased UBE3A levels in the small population of immature neurons that biallelically express UBE3A, and/or to the cytotoxicity of the compound on these cells.

We have now emphasized these points in the manuscript.

c) I am surprised that the authors have found such a great compound in the 2800 compound library. In the future, using this great mouse line it may be worth considering the FDA approved libraries or the much larger billion compound libraries. In the absence of this, the authors should discuss the reason for using this library. What is its complexity? What types of compounds are there in the library that increases confidence of broad survey of compound types with unique features?

We would love to be able to perform a screen on millions to billions of compounds, but such screens are only feasible for dividing cell types. Because *Ube3a* is only imprinted in mature neurons, we can only perform the screen in primary neurons, greatly increasing the cost and time expenditures, thus limiting the number of compounds we can realistically screen. Accordingly, we carefully chose a chemogenomic library, in this case provided by Pfizer through an exclusive agreement, due to its distinct advantages in expediting drug discovery endeavors. This library was meticulously designed to encompass a broad array of pharmacological space, comprising compounds with well-characterized pharmacology suitable for phenotypic screens⁵. Each compound within the Pfizer chemogenomic library was selected based on its potency against annotated biological targets and its broad selectivity across the proteome. This strategic approach ensures that hits obtained from screening not only exhibit efficacy against their designated targets, but also offer the potential for elucidating novel mechanisms of action and therapeutic synergies. Moreover, the selection process includes stringent criteria such as selectivity, permeability, solubility, and cytotoxicity, ensuring the suitability of these compounds for cell-based screening.

Overall, the Pfizer chemogenomic library provides a systematic and comprehensive framework for phenotypic screening, facilitating the discovery of promising therapeutic candidates and the exploration of intricate biological mechanisms. However, we acknowledge the importance of continuing screening efforts with other libraries, including FDA-approved ones, to further enrich our exploration of novel unsilencers of paternal *Ube3a*.

We have added a short description, together with a reference to the review article discussing the chemogenomic small molecule library, describing the library and contextualizing our screening results.

d) Please indicate time of treatment for figure 1b.

We have added the time of treatment (72 hours) to the **Figure 1** legend.

e) Was Figure 1b accomplished through live imaging or imaging of fixed and stained cells? If fixed and stained, include these details in your figure 1a.

All immunofluorescent images presented in the paper depict fixed cells stained with antibodies, as outlined in the Methods section. For clarity, we now explicitly mention in the figure legend for **Figure 1** that the data were generated from fixed immunolabeled cells.

Figure 2:

a) Please include some stats on these graphs?

The dose-response graphs in **Figure 2** represent data from runs in quadruplicate, showing means and standard error. Dose-response curves were generated using four-parameter logistic sigmoidal curve fitting using GraphPad Prism software version 8.0, as stated in the methods section. Following the standard in the field, these dose-response curves do not include statistical analysis, and their importance lies in being able to establish pharmacological profiles, including EC_{50} and E_{MAX} values. Note that these values cannot be recorded without being able to establish sufficient sigmoidal dose-response curves.

b) I was not clear on what the authors mean by “Viability (% of DMSO)”. Why do the cells die with more paternal UBE3A? Make the discussion of this figure as clear as possible.

The cells are unlikely to exhibit toxicity due to higher concentrations of UBE3A, as we have previously shown that high gene dosage of *Ube3a* is well tolerated⁶. Rather, the cells are likely dying from CDK2/5-related toxicity or an off-target effect of (S)-PHA533533 or its analogs (a focus of current investigation).

A cytotoxicity dose-response curve (presented with a red line) shows the relationship between the concentration of a compound and its effect on cell viability or survival. This curve illustrates how the percentage of viable cells changes in response to increasing doses of the compound (compared to DMSO-treated control wells, which are presented as % of DMSO), providing information about the compound’s toxicity profile and its potential to cause cell death at different concentrations.

For clarity, we have now added an efficacy measure (E_{MAX}) of all compounds to the table in **Figure 2a** (presented as % patUBE3A-YFP+), and corresponding text, detailing how the efficacy curves were generated. Specifically, we highlight that only data points where at least 90% of neurons remained viable compared to the DMSO control wells were included. Concentrations above which >10% of neurons were killed compared to DMSO-treated wells were excluded from E_{MAX} analysis, to focus on results achievable in fully viable neurons.

c) PHA5 should be tested in a non-neuronal cell? What happens to ube3a levels in the non-neuronal cells that express the paternal allele and are treated with PHA5?

We treated HEK293T cells that express *UBE3A* biallelically with 1 μ M (S)-PHA533533 for 48 hours. Our results showed no significant change in the *UBE3A* mRNA or protein levels following (S)-PHA533533 treatment in HEK293T cells (new **Supplementary Figure 4a-c**).

We now provide these data in our Results and **Supplementary Figure 4**.

d) Show total levels of UBE3A in wild type neurons treated with PHA5.

We have added data demonstrating that both (S)-PHA533533 and topotecan treatments increase *Ube3a* mRNA and UBE3A protein levels in wild-type neurons. Thus, demonstrating that the unsilencing of paternal *Ube3a* in neurons is independent of the presence of YFP reporter (*Ube3a^{m+/p^{YFP}}*) or the mutation in the maternal *Ube3a* allele (*Ube3a^{m-/p+}*) (see new **Supplementary Figure 2a-b**).

We now provide these data in our Results and **Supplementary Figure 2**.

e) What is the time course of treatment for these assays?

We have specified in the figure legend that all drug treatments were conducted for 72 hours.

f) Counting NeuN and Dapi, I do not believe is definitive for viability measurements. It is possible that neurons are detaching from the plate and yet still alive. Maybe change the title to percent of cells. One thought is whether the decrease in cell density is giving rise to the increase in % neurons that have UBE3A-YFP.

We appreciate your concern regarding the method used to assess viability. While it is true that counting NeuN and DAPI double-positive cells may not be definitive for viability measurements in all contexts, it is important to note that detached neurons would not contribute to this count as they would not be present in the well at the time of analysis (as the detached cells are removed during washes, for example). We trust that our clarification provided here, along with additional details specified in the figure legend and results section regarding the data presented in **Figure 2**, sufficiently justifies the suitability of expressing viability as a percentage of DMSO-treated neurons.

As stated in our answer above (b), we specifically avoided measuring the increase in paternal UBE3A-YFP expression at concentrations where cells showed signs of toxicity toward drug treatment, therefore only data points where at least 90% of neurons remained viable compared to the DMSO control wells were included. We add this information in the text describing **Figure 2** and in the figure legend.

Figure 3:

a) Put molecular weight markers on each immunoblot. I was having trouble telling whether I am looking at endogenous UBE3A or UBE3A-YFP. Is this full length UBE3A?

Western blot data presented in **Figure 3g** shows UBE3A expression in mouse primary neurons derived from wild-type or AS (*Ube3a^{m-/p+}*) mice.

We have added molecular weight marker bands to immunoblots on all figures (**Figure 3, Figure 4, and Supplementary Figures 2 and 3**).

b) What is the mRNA level of *ube3a* in WT neurons treated with PHA5?

We have added data demonstrating that both (S)-PHA533533 and topotecan treatments increase the *Ube3a* mRNA and protein levels in wild-type neurons. Thus, demonstrating that the unsilencing of paternal *Ube3a* in neurons is independent of the presence of YFP reporter (*Ube3a^{m+/p^{YFP}}*) or the mutation in the maternal *Ube3a* allele (*Ube3a^{m-/p+}*) (see new **Supplementary Figure 2a-b**).

We now provide these data in our Results and **Supplementary Figure 2**.

c) What is the mRNA level of ube3a in a non-neuronal cell treated with PHA5?

We treated HEK293T cells that express *UBE3A* biallelically with 1 μ M (S)-PHA533533 for 48 hours. Our results showed no significant change in the *UBE3A* mRNA or protein levels following (S)-PHA533533 treatment in HEK293T cells (new **Supplementary Figure 4a-c**).

We now provide these data in our Results and **Supplementary Figure 4**.

Figure 4:

a) Put molecular weight markers on each immunoblot. I was having trouble telling whether I am looking at endogenous UBE3A or UBE3A-YFP. Is this full length UBE3A?

Western blot data presented in **Figure 4a, c, e** shows UBE3A-YFP expression in mouse primary neurons derived from wild-type or paternal *Ube3a-YFP* reporter mice (*Ube3a^{m+/pYFP}*).

We have added molecular weight marker bands to immunoblots on all figures (**Figure 3, Figure 4, and Supplementary Figures 2 and 3**).

b) What happens to total non-YFP labelled maternal ube3a in the PHA5 treated neurons in this figure? Please show.

We have added data demonstrating that both (S)-PHA533533 and topotecan treatments increase the *Ube3a* mRNA and protein levels in wild-type neurons. Thus, demonstrating that the unsilencing of paternal *Ube3a* in neurons is independent of the presence of YFP reporter (*Ube3a^{m+/pYFP}*) or the mutation in the maternal *Ube3a* allele (*Ube3a^{m-/p+}*) (see new **Supplementary Figure 2a-b**).

We now provide these data in our Results and **Supplementary Figure 2**.

Figure 5:

a) I was having trouble understanding the data in 5A compared to 5B-5D. The differences by fluorescence appear to be far greater than what is quantified in 5A. The fluorescence does not appear to be a 50% difference. It appears to be a far greater difference. This may be a small point; however, the authors should discuss it and make every effort to explain why they may see these differences compared to their immunoblots and mRNA measurements.

We are hesitant to compare the quantitative PCR measurements to the immunofluorescent images. The apparent differences observed between the qPCR results (**Figure 5a**) and the immunohistochemistry findings (**Figure 5b-d**) can be attributed to several factors. First, it is important to note that qPCR measures gene expression levels at a bulk tissue level (e.g. – neurons and glia, whose *Ube3a* expression is unaffected by (S)-PHA533533 as they lack the *Ube3a-ATS*), providing an overall assessment of mRNA abundance within the sample. On the other hand, immunohistochemistry allows for the visualization and quantification of protein expression at a cellular level, providing spatial information regarding the localization and intensity of protein expression within individual cells – this draws attention to the high levels of UBE3A protein in neurons, as glia express lower levels of UBE3A.

Furthermore, the observed differences may also reflect the temporal dynamics of gene expression and protein synthesis, where changes in mRNA levels may precede or lag changes in protein abundance. Additionally, post-transcriptional regulatory mechanisms, such as protein stability, translation efficiency, and post-translational modifications, can further modulate the relationship

between mRNA and protein levels. Thus, future studies are needed to carefully map the time course of upregulation of *Ube3a* mRNA and UBE3A protein following (S)-PHA533533 application.

Figure 6:

a) What did these neurons look like? The authors should show other neuronal markers and how they change with these treatments.

In new **Supplemental Figure 3**, we add staining for the neuronal marker NeuN. We also add that the neurons appeared normal/healthy until we reached higher drug concentrations. However, it is important to note that these cells are not typical neurons. As artificial cell lines, they were primarily used to assess *UBE3A-ATS* reduction and *UBE3A* restoration, and are not ideal lines for evaluating effects of neuronal function.

b) Would be great to see an immunoblot of these treated samples? c) Did the UBE3A protein level go up?

In new **Supplementary Figure 3**, we have added western blot data demonstrating that (S)-PHA533533 treatment increases UBE3A protein levels in 10-week-old iPSC-derived neurons from an individual with Angelman syndrome (ASdel1-0). In addition, we also show that UBE3A levels are increased in ASdel1-0 cells using immunocytochemistry.

d) How about viability measurements in these cells?

We now clarify our viability measurements. We have specified that the CC_{50} value was calculated based on a viability assessment using Ct values of a housekeeping gene *GAPDH*, and this is described in the text describing data in **Figure 6** and in the figure legend.

Overall:

This is an interesting study. A model of how this compound may be working would be a nice addition to the study. The major benefit of this study is the identification of the compound. Using the compound in future studies and sharing with AS community may have major positive implications for AS research and possibly the patient population.

We appreciate the suggestion that a model of how this compound may be working would be useful, but at this time the mechanism of action is unknown and any such model would be highly speculative. Thus, we prefer not to include a model at this time. However, our major research efforts now lie in elucidating the mechanism by which (S)-PHA533533 unsilences paternal *UBE3A*, as this will help our translational efforts.

References

1. Perry, R. B.-T. & Ulitsky, I. The functions of long noncoding RNAs in development and stem cells. *Development* **143**, 3882–3894 (2016).
2. Hsiao, J. S. *et al.* A bipartite boundary element restricts UBE3A imprinting to mature neurons. *Proc. Natl. Acad. Sci. U. S. A.* **116**, 2181–2186 (2019).
3. Sen, D., Voulgaropoulos, A., Drobna, Z. & Keung, A. J. Human Cerebral Organoids Reveal Early Spatiotemporal Dynamics and Pharmacological Responses of UBE3A. *Stem Cell Rep.* **15**, 845–854 (2020).
4. Huang, H.-S. *et al.* Topoisomerase inhibitors unsilence the dormant allele of Ube3a in neurons. *Nature* **481**, 185–189 (2011).
5. Jones, L. H. & Bunnage, M. E. Applications of chemogenomic library screening in drug discovery. *Nat. Rev. Drug Discov.* **16**, 285–296 (2017).
6. Punt, A. M. *et al.* Molecular and behavioral consequences of *Ube3a* gene overdosage in mice. *JCI Insight* **7**, (2022).

REVIEWERS' COMMENTS

Reviewer #1 (Remarks to the Author):

The authors have addressed my concerns. The article is appropriate for publication.

Reviewer #2 (Remarks to the Author):

The authors have addressed all my previous comments very clearly.

Reviewer #3 (Remarks to the Author):

Based on the reported findings, treating neurons with (S)-PHA533533 leads to higher levels of paternally expressed ube3a in the maternally null animals. Based on the findings, it remains possible that (S)-PHA533533 could lead to a viable means for increasing ube3a expression in patients with mutant or no ube3a. Whether (S)-PHA533533 will serve as a treatment for AS remains to be determined. Moreover, although investigated to some extent in this study, whether (S)-PHA533533 has off target negative impact is not known at this time. The work is rigorous and supports the claims made in the study.